# Evaluation of the methane paradox in four adjacent pre-alpine lakes across a trophic gradient

César Ordóñez [1] ✉, Tonya DelSontro [1,2] ✉, Timon Langenegger[1], Daphne Donis [1], Ena L. Suarez[1] & Daniel F. McGinnis [1] ✉

Contrasting the paradigm that methane is only produced in anoxic conditions, recent discoveries show that oxic methane production (OMP, aka the methane paradox) occurs in oxygenated surface waters worldwide. OMP drivers and their contribution to global methane emissions, however, are not well constrained. In four adjacent pre-alpine lakes, we determine the net methane production rates in oxic surface waters using two mass balance approaches, accounting for methane sources and sinks. We find that OMP occurs in three out of four studied lakes, often as the dominant source of diffusive methane emissions. Correlations of net methane production versus chlorophyll-*a*, Secchi and surface mixed layer depths suggest a link with photosynthesis and provides an empirical upscaling approach. As OMP is a methane source in direct contact with the atmosphere, a better understanding of its extent and drivers is necessary to constrain the atmospheric methane contribution by inland waters.

The widely reported methane ($CH_4$) oversaturation in surface oxic waters in oceans[1] and lakes (also referred to as the methane paradox; Tang et al.[2]) contrasts with the current understanding that biogenic $CH_4$ formation occurs exclusively under anoxic conditions[3]. Methane production in oxic conditions (also called oxic methane production or OMP) has been reported for an increasing number of lakes[2,4–8]. While recent studies have shown that OMP may have contributed up to 80% of lake-diffusive $CH_4$ emissions[7,8], other researchers suggest that $CH_4$ produced in anoxic littoral sediments is enough to resolve the $CH_4$ paradox[9–12]. Thus, the drivers and OMP contribution to global lake $CH_4$ emissions remain unclear.

Atmospheric $CH_4$ concentrations have more than doubled since the onset of the industrial era[13]. Although $CH_4$ is less abundant in the atmosphere than carbon dioxide ($CO_2$), the global warming potential (GWP) of $CH_4$ is ~80 times higher than $CO_2$ over a 20-year period[14]. Its GWP combined with its ~12-year lifetime means that reducing $CH_4$ emissions is a priority for mitigating climate change[15]. Lakes represent ~25% of natural $CH_4$ atmospheric sources, but large uncertainties

remain about the contribution of internal sources and sinks[16–18]. Methane in lakes can be emitted to the atmosphere through bubbles (ebullition) and diffusive fluxes at the air-water interface (AWI)[17]. Ebullition is driven by high $CH_4$ production rates in anoxic sediments[19] and the diffusive fluxes at the AWI are driven by $CH_4$ concentrations and turbulence in surface waters[20]. As OMP occurs in surface waters, the $CH_4$ can be quickly emitted to the atmosphere[2].

Several mechanisms have been proposed for OMP[1,2,4,5,21–23], however, recent studies have shown correlations between $CH_4$, oxygen, and phytoplankton concentrations[4,6], suggesting a direct role of phytoplankton in OMP[6,12,24,25]. Although the mechanisms remain unclear, OMP has been shown to follow light-dark cycles in different phytoplankton cultures[24,25]. It is likely that multiple pathways produce $CH_4$ in oxic lake environments, and that these may vary from lake-to-lake and seasonally based on trophic properties and light conditions.

OMP rates have been reported using different methodologies, such as laboratory and in-situ water incubations[4,24,25], in-lake mesocosms[5,8], a physical lateral transport model[26], and lake mass

[1]Aquatic Physics Group, Department F.-A. Forel for Environmental and Aquatic Sciences (DEFSE), Faculty of Science, University of Geneva, Uni Carl Vogt, 66 Boulevard Carl-Vogt, 1211 Geneva, Switzerland. [2]Now at Department of Earth and Environmental Sciences, University of Waterloo, Waterloo, ON, Canada. ✉e-mail: cesar.ordonez@unige.ch; tonya.delsontro@uwaterloo.ca; daniel.mcginnis@unige.ch

balances[7,8]. OMP has also been reported from different freshwater environments, including both temperate[27] and polar regions[28], high altitude lakes (above 2500 m.a.s.l)[29], tropical latitude lakes[12], and across a range of trophic states[27,30]. While these studies show the occurrence of OMP in lakes across geographic and trophic gradients, OMP has not been investigated in pre-alpine lakes.

Pre-alpine lakes (from 1300 to 2000 m.a.s.l) are atmospheric $CH_4$ sources[31] and, with air temperature increasing two times faster in the European Alps than the global mean[32], these lakes are disproportionately experiencing climate change[33,34]. Such an increase in air temperature can induce limnological change in pre-alpine lakes with implications for aquatic $CH_4$ emissions[33–35] such as: (1) a longer ice-free season that allows $CH_4$ to be emitted for a longer period[36]; (2) an increase in surface water temperature that enhances littoral production rates of $CH_4$[37]; and (3) a longer stratified season that allows for more $CH_4$ accumulation in the hypolimnion[38]. These impacts will differ across lakes depending on the light regime and trophic state[39]; therefore, the precise impact of climate change on the $CH_4$ budget in pre-alpine lakes needs further investigation.

This study focuses on four adjacent Swiss pre-alpine lakes under identical climate forcing but with different trophic states. The net $CH_4$ production rate ($P_{net}$, Fig. 1) is defined as the balance between OMP (adds $CH_4$) and $CH_4$ oxidation (MOx, removes $CH_4$) in the surface mixed layer (SML)[40]. $P_{net}$ in the SML was estimated using two independent mass balance approaches: a 0-D full-scale mass balance following Donis et al.[7] and a 1-D lateral transport model adapted from Peeters et al.[9]. In the latter, we included two additional terms—diffusive $CH_4$ flux across the thermocline and $CH_4$ bubble dissolution (Fig. 1). In both models, we included all $CH_4$ sources in the SML and the loss to the atmosphere to determine $P_{net}$ (Fig. 1). Note that MOx rates are not necessary to estimate $P_{net}$ (Methods), but would be required to calculate OMP. Here, we determine $P_{net}$ ($P_{net}$ = OMP − MOx) as this is the component that contributes to the diffusive emissions.

In previous studies, the two models have shown contradictory results mainly due to the use of literature parameterizations to estimate surface diffusive emissions[7–9,41,42]. Instead, we used direct flux measurements from floating chambers and found the two models agree well with each other ($R^2 = 0.97$). The results indicate that $P_{net}$ contributes between 30 and 90% to diffusive emissions during the stratified period of pre-alpine lakes with different trophic states. Moreover, we performed a sensitivity analysis using five diffusive flux literature parameterizations and surface flux measurements to analyze the impact that modeled versus measured atmospheric diffusive fluxes have on $P_{net}$ estimation. Finally, we present a $P_{net}$ upscaling approach based on chlorophyll-$a$ concentrations (Chl$a$), light penetration, and SML depth. Ultimately, our findings highlight the need for $P_{net}$ to be included in $CH_4$ lake budgets and for more research to understand OMP($P_{net}$) drivers and its response to climate change.

## Results

### Study sites

The four pre-alpine lakes studied - Lac de Bretaye (BRE), Lac Noir (NOI), Lac des Chavonnes (CHA), and Lac Lioson (LIO)—are located between 1650 to 1850 m.a.s.l in the Swiss Alps and are eutrophic, meso/eutrophic, mesotrophic and oligotrophic, respectively (Supplementary Table 1 and 2). NOI and BRE are small lakes with a maximum depth of ~9 m, while CHA and LIO have a maximum depth of ~28 m (Supplementary Fig. 1 and Table 1). Throughout the three sampling campaigns (June 2018, September 2018, and July 2019), the surface waters of all four lakes were oxic and oversaturated in $CH_4$ (Table 1 and Supplementary Fig. 2). Temperature and $CH_4$ concentration profiles at the deepest point of the lakes showed that all the lakes were stratified (SML thickness 1–6 m, Table 1 and Supplementary Fig. 2). Secchi depths ($Z_s$), nutrients and Chl$a$ concentrations reflect the trophic gradient of the study lakes (Table 1).

### Surface methane concentration and isotopic signature

Surface $CH_4$ concentrations and their stable isotopic signatures ($\delta^{13}C_{CH_4}$) were measured at the deepest point of each lake (Supplementary Fig. 2) and along a transect from shore to shore to resolve their spatial variability in the SML (Fig. 2 and Supplementary Fig. 3). All four lakes were oversaturated with $CH_4$, with near the shore values $33 \pm 32\%$ higher than in the center (all results are reported in mean ± 1 standard deviation, SD, unless otherwise indicated), although only 40% of the time this difference was significant (Table 1). The eutrophic lakes BRE and NOI, on average, had one order of magnitude higher surface concentrations ($3.13 \pm 2.09$ mmol m$^{-3}$) than the oligo/mesotrophic lakes LIO and CHA ($0.15 \pm 0.13$ mmol m$^{-3}$) (Table 1).

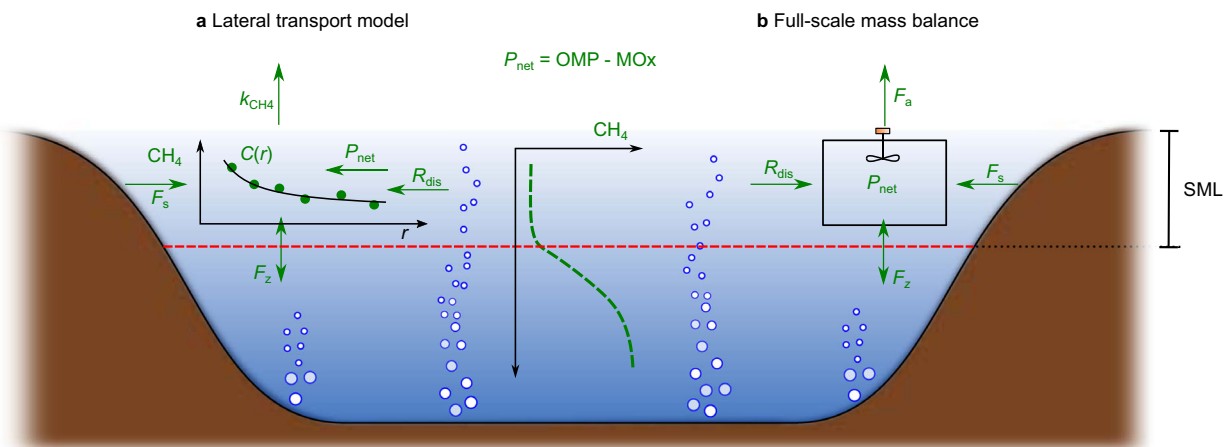

**a** Lateral transport model

**b** Full-scale mass balance

$P_{net}$ = OMP - MOx

**Fig. 1 | Conceptual schematic of the $CH_4$ budget components in the surface mixed layer (SML) and methodological approaches.** $CH_4$ mass balance components: diffusive $CH_4$ emissions to the atmosphere ($F_a$), vertical transport ($F_z$), bubble dissolution ($R_{dis}$), littoral sediment flux ($F_s$). The net $CH_4$ production rate ($P_{net}$) in the SML is estimated using a 1-D lateral transport model and a 0-D full-scale mass balance in **a** and **b**, respectively. $P_{net}$ is the balance between oxic $CH_4$ production (OMP, adds $CH_4$) and $CH_4$ oxidation (MOx, removes $CH_4$). The full-scale mass balance assumes the SML as a well-mixed reactor where each component is based on measured values. The lateral transport model also used in situ measurements but estimates the diffusive flux to the atmosphere using the mass transfer coefficient ($k_{CH4}$) and $P_{net}$ rates are obtained by finding the simulated transect $CH_4$ concentrations ($C(r)$) that best-fit the measured $CH_4$ concentrations.

**Table 1 | General characteristics of surface waters across the studied lakes**

| Lake | Date | $CH_4$ (mmol m$^{-3}$) | $\delta^{13}C_{CH_4}$ (‰) | $\Delta CH_4/CH_{4shore}$ (%) | Secchi depth (m) | $H_{SML}$ (m) | Chl$a$ (mg m$^{-3}$) | DIN (mg m$^{-3}$) | DP (mg m$^{-3}$) |
|---|---|---|---|---|---|---|---|---|---|
| Bretaye | June 2018 | 6.7 ± 2.3 | −52.0 | 54 | 3.7 | 1.3 | 3.01 | 18 | 9.0 |
| | Sept 2018 | 3.5 ± 0.5 | −38.0 | 22* | 3.0 | 5.2 | 4.08 | 29 | 7.3 |
| | July 2019 | 2.8 ± 1.6 | −48.8 | 4 | 4.7 | 2.6 | 4.05 | 4 | 57 |
| Noir | June 2018 | 1.4 ± 0.1 | −54.5 | 18* | 2.8 | 0.9 | 8.81 | 18 | 2.3 |
| | Sept 2018 | 1.8 ± 0.4 | −45.5 | 19 | 6.1 | 5.4 | 4.71 | 13 | 2.7 |
| | July 2019 | 3.9 ± 0.3 | −49.9 | 23 | 3.8 | 1.9 | 8.48 | BD | BD |
| Chavonnes | June 2018 | 0.1 ± 0.1 | −62.3 | 59* | 4.6 | 1.3 | 3.73 | 235 | 2.0 |
| | Sept 2018 | 0.2 ± 0.1 | −62.4 | 22 | 5.2 | 4.6 | 2.51 | 167 | 1.0 |
| | July 2019 | 0.1 ± 0.0 | −61.2 | 120 | 3.8 | 2.0 | 5.02 | 189 | BD |
| Lioson | June 2018 | 0.1 ± 0.0 | −50.9 | 33* | 9.0 | 0.9 | 1.52 | 126 | 2.0 |
| | Sept 2018 | 0.4 ± 0.6 | −50.1 | 12* | 10.5 | 6.1 | 3.01 | 45 | 1.0 |
| | July 2019 | 0.2 ± 0.2 | −54.0 | 14 | 5.5 | 2.2 | 4.64 | 71 | BD |

Spatial average of surface $CH_4$ concentration (mean ±1 SD) and its stable isotopic signature ($\delta^{13}C_{CH_4}$) along each transect. Average dissolved inorganic nitrogen (DIN), chlorophyll-$a$ concentration (Chl$a$), and dissolved phosphorus (DP) concentrations in the surface mixed layer (SML). The Secchi and SML depth ($H_{SML}$) at each sampling campaign in each lake. $\Delta CH_4/CH_{4shore}$ is the percentage difference between the $CH_4$ concentration at shore and the center. The values marked with * signify that there is a significant difference between shore and center as determined with an ANOVA analysis.

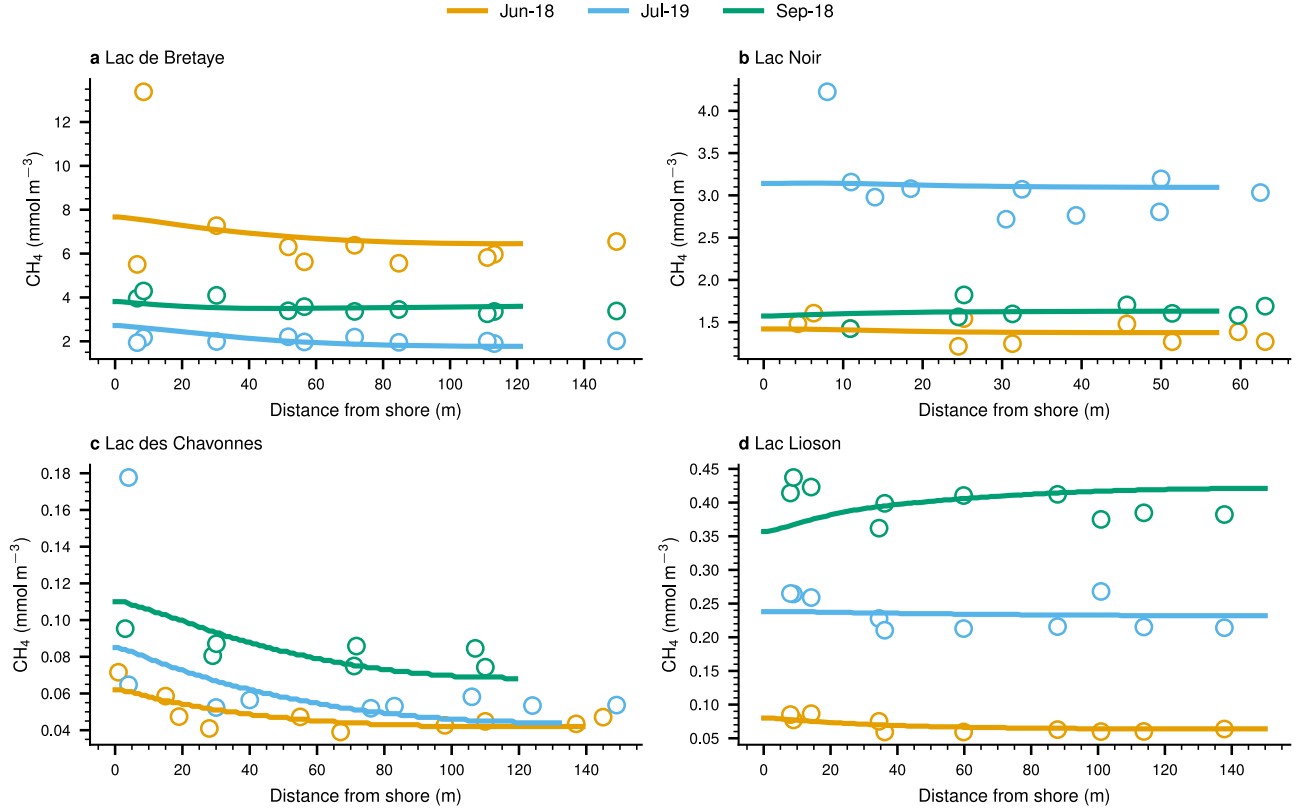

**Fig. 2 | Surface $CH_4$ concentrations along the transects sampled in each lake.** **a** Lac de Bretaye, **b** Lac Noir, **c** Lac des Chavonnes, and **d** Lac Lioson. Lines represent the $CH_4$ concentration simulated using the lateral transport model and dots are the measured values. Since the lateral transport model assumes that the $CH_4$ concentrations in the SML are radially symmetric, the concentrations are shown from shore to center. The bathymetry profile along the transects is shown in Supplementary Fig. 4.

During two transects surveys in NOI (July-19 and Sep-18), one in BRE (Jun-18) and one in CHA (Jul-19) we observed one concentration data point close to the shore that was between 36 and 160% higher than the overall average transect concentration. In NOI and BRE, the presence of macrophytes could have decreased the horizontal dispersion, produce $CH_4$[43] and/or decreased surface $CH_4$ emissions, resulting in near-shore $CH_4$ accumulation not accounted for in the lateral transport model. Since we measured low $CH_4$ concentrations in CHA, any

disturbance in the lake sediment could have caused an increase of $CH_4$ near the shore.

The spatially averaged $\delta^{13}C_{CH_4}$ signature ranged between −62 ± −38‰ (Table 1). Isotopically enriched $CH_4$ ($\delta^{13}C_{CH_4} \sim -40$‰) was observed at the end of summer in the SML of the eutrophic lakes, while in the oligotrophic lakes $\delta^{13}C_{CH_4}$ was relatively consistent between sampling dates (Supplementary Table 3). Rather constant $\delta^{13}C_{CH_4}$ values were observed along the transect for most of the lakes, except

**Table 2 | Inputs for the lateral transport model and full-scale mass balance in the surface mixed layer (SML) (mean ± SD)**

| Lake | Date | $K_H$ (m² d⁻¹) | $C_{hyp}$ (mmol m⁻³) | $K_z$ (10⁻⁶ m² s⁻¹) | $\bar{k}_{CH_4}$ (m d⁻¹) | $F_s$ (mmol m⁻² d⁻¹) | $F_a$ (mmol m⁻² d⁻¹) | $F_z$ (mmol m⁻² d⁻¹) | $R_{dis}$ (µmol m⁻³ d⁻¹) |
|---|---|---|---|---|---|---|---|---|---|
| Bretaye | June 2018 | 2034 | 4.0 | 4.09 | 0.67 | 8.3 ± 6.7 (n = 3) | 4.6 ± 1.8 | 0.5 ± 0.3 | 50.6 ± 10.2 |
|  | Sept 2018 |  | 161.8 | 0.96 | 1.00 |  | 3.7 ± 1.5 | 13.3 ± 7.7 | 34.9 ± 9 |
|  | July 2019 |  | 2.3 | 0.94 | 2.12 |  | 3.7 ± 1.6 | 0.02 ± 0.01 | 42.7 ± 11.3 |
| Noir | June 2018 | 903 | 1.3 | 0.91 | 1.75 | 1.5 ± 0.3 (n = 4) | 2.4 ± 0.8 | 0.03 ± 0.02 | 17.2 ± 1.6 |
|  | Sept 2018 |  | 13.7 | 30.1 | 1.48 |  | 2.2 ± 1.0 | 3.1 ± 1.8 | 24.7 ± 8.1 |
|  | July 2019 |  | 2.3 | 0.07 | 0.69 |  | 2.9 ± 1.7 | −0.01 ± 0 | 17.0 ± 1.7 |
| Chavonnes | June 2018 | 2366 | 0.1 | 14.14 | 2.23 | 0.4 ± 0.4 (n = 3) | 0.1 ± 0.02 | −0.1 ± 0.03 | 0 ± 0 |
|  | Sept 2018 | 2004 | 0.1 | 0.74 | 1.49 |  | 0.2 ± 0.1 | 0.0 ± 0 | 0 ± 0 |
|  | July 2019 | 2246 | 0.4 | 1.02 | 1.12 |  | 0.1 ± 0.1 | 0.03 ± 0.02 | 0 ± 0 |
| Lioson | June 2018 | 2564 | 0.1 | 0.89 | 2.22 | 0.3 ± 0.1 (n = 3) | 0.2 ± 0.04 | 0 ± 0 | 0 ± 0 |
|  | Sept 2018 |  | 0.6 | 0.03 | 3.30 |  | 1.2 ± 0.6 | 0 ± 0 | 0 ± 0 |
|  | July 2019 |  | 0.3 | 4.80 | 1.29 |  | 0.4 ± 0.2 | 0.01 ± 0.01 | 0 ± 0 |

$K_H$ is the horizontal dispersion coefficient, $C_{hyp}$ is the $CH_4$ concentration 1 m below the SML, $K_z$ is the vertical diffusivity at the base of the epilimnion and $\bar{k}_{CH_4}$ is the average chamber-based mass transfer coefficient. $F_s$, $F_a$, $F_z$, and $R_{dis}$ are the littoral sediment flux, surface diffusive emissions, vertical flux at the base of the epilimnion, and the bubble dissolution rate in the SML, respectively.

for CHA in June 2018 when lighter $\delta^{13}C_{CH_4}$ was observed at the shore (∼−65‰) than in the center of the lake (∼−60‰) (Supplementary Fig. 3).

**Diffusive CH₄ emissions to the atmosphere**
Diffusive CH₄ emissions ($F_a$) at the air-water interface (AWI) were measured in each lake using a floating chamber[44] at the deepest point of the lake and along the transects. Average surface fluxes measured in the eutrophic lakes (NOI and BRE, 3.24 ± 0.88 mmol m⁻² d⁻¹) were an order of magnitude higher than in the oligotrophic lakes (LIO and CHA, 0.29 ± 0.43 mmol m⁻² d⁻¹). Surface diffusive fluxes of CH₄ remained relatively similar between sampling dates in each lake (Table 2).

Several parameterizations have been proposed for the mass transfer coefficient ($k_{600}$) used along with CH₄ concentrations to estimate atmospheric diffusive emissions (Klaus & Vachon[45] and references therein). We compared CH₄ mass transfer coefficients based on our chamber flux data ($k_{600}^{cb}$) to five $k_{600}$ parameterizations: CC98 based on Cole & Caraco[46]; MA10-NP (negative buoyancy), MA10-MB (mixed buoyancy), and MA10-PB (positive buoyancy) based on MacIntyre et al.[20]; and VP13 based on Vachon & Prairie[47] (Supplementary Fig. 5). These parameterizations weakly correlated with $k_{600}^{cb}$ ($R^2$ = [0.01–0.037]; Supplementary Fig. 6) and underestimated $k_{600}^{cb}$ (Mean Normalized Bias (MNB) = [16–81%]) (Supplementary Fig. 6). The best agreement was found with MA10-NB which is based on convective mixing ($R^2$ = [0.01–0.37], RMSE = [0.63–4.65 m d⁻¹], MNB = [16–57%]; Supplementary Fig. 6).

**Diffusive CH₄ fluxes from littoral sediments**
Diffusive CH₄ fluxes at the sediment-water interface (SWI) in the littoral zone ($F_s$) were estimated using benthic chambers and porewater measurements of dissolved CH₄ (Supplementary Fig. 7 and Supplementary Table 4). The highest average littoral sediment flux was found in eutrophic BRE (8.3 ± 6.7 mmol m⁻² d⁻¹), followed by NOI (eutrophic), CHA (mesotrophic) and LIO (oligotrophic) with the lowest value (0.3 ± 0.1 mmol m⁻² d⁻¹) (Table 2). $\delta^{13}C_{CH_4}$ in the upper part of the sediments ranged between −66 and −48‰ (Supplementary Table 3). Littoral sediment was ∼20% isotopically less enriched than the surface waters of NOI and BRE but similar for CHA (−60‰, Supplementary Table 3 and Fig. 7). No porewater measurements were performed in LIO due to the rocky nature of the littoral sediments (Methods).

**CH₄ ebullition rates and bubble dissolution**
CH₄ ebullition rates at the SWI were estimated using the gas composition of bubbles collected during each sampling campaign, the CH₄ fluxes measured at the SWI (Supplementary Table 4), and modeling the

dissolved porewater gas concentration in the sediments following Langenegger et al.[19]. Bubble dissolution rates in the SML ($R_{dis}$) were obtained using a discrete bubble model[48] (Methods). The spatially averaged ebullitive fluxes ($F_{eb}$) for BRE and NOI (1.14 and 0.43 mmol m⁻² d⁻¹, respectively), resulted in bubble dissolution rates between 17 and 51 µmol m⁻³ d⁻¹ (Table 2). Ebullition was not detected in CHA and LIO.

**Vertical diffusive fluxes from/to the epilimnion**
The vertical transport from/to the epilimnion ($F_z$) is determined with Fick's 1st Law using the turbulent vertical diffusivity ($K_z$) and concentration gradients at the base of the epilimnion. $K_z$ values at the top of the thermocline ranged between 0.03 and 14.4 × 10⁻⁶ m² s⁻¹ (Table 2). In all lakes, $F_z$ was typically low (−0.1–0.5 mmol m⁻² d⁻¹), except in BRE and NOI at the end of the summer when fluxes were 13.3 and 3.1 mmol m⁻² d⁻¹, respectively.

**Horizontal dispersion**
In the lateral transport model, we estimated the horizontal dispersion coefficient ($K_H$) for each lake using Peeters & Hofmann[49] parametrization (Methods). Water level fluctuations were minimal in BRE, NOI, and LOI (± 1 m). In CHA, the highest water level was observed at the beginning of summer after ice-off and slowly decreased during the summer by about 4 m (Supplementary Fig. 8), which changed the length scale ($L$) and thus $K_H$ (Eq. (4)). The calculated $K_H$ values were 2034, 903, and 2564 m² d⁻¹ for BRE, NOI, and LIO, respectively, and ranged between 2004–2366 m² d⁻¹ for CHA (Table 2).

**Surface mass balances**
The full-scale mass balance (0-D) proposed by Donis et al.[7] (Eq. (1)) and a modified version of the lateral transport model (1-D) proposed by Peeters et al.[9] (Eq. (2)) were used to determine $P_{net}$ in the SML of each lake and campaign based on the input values listed in Table 2. $P_{net}$ is the net result of OMP and MOx (i.e., $P_{net}$ = OMP − MOx), which adds and removes CH₄ to the SML, respectively. Thus, when $P_{net}$ is positive the true OMP rate is actually higher than $P_{net}$.

Despite the different modeling approaches and underlying assumptions, the $P_{net}$ rates calculated with both models under steady-state conditions correlated well with each other (Supplementary Fig. 9, $R^2$ = 0.97). Monte Carlo simulations were applied to assess uncertainties using all sources and sinks in both models during the stratified period (Methods). The average $P_{net}$ rates for the three sampling dates were 305, 1504, 22, and 246 µmol m⁻³ d⁻¹ for BRE, NOI, CHA, and LIO, respectively (Fig. 3). On average, $P_{net}$ rates in eutrophic lakes (BRE and

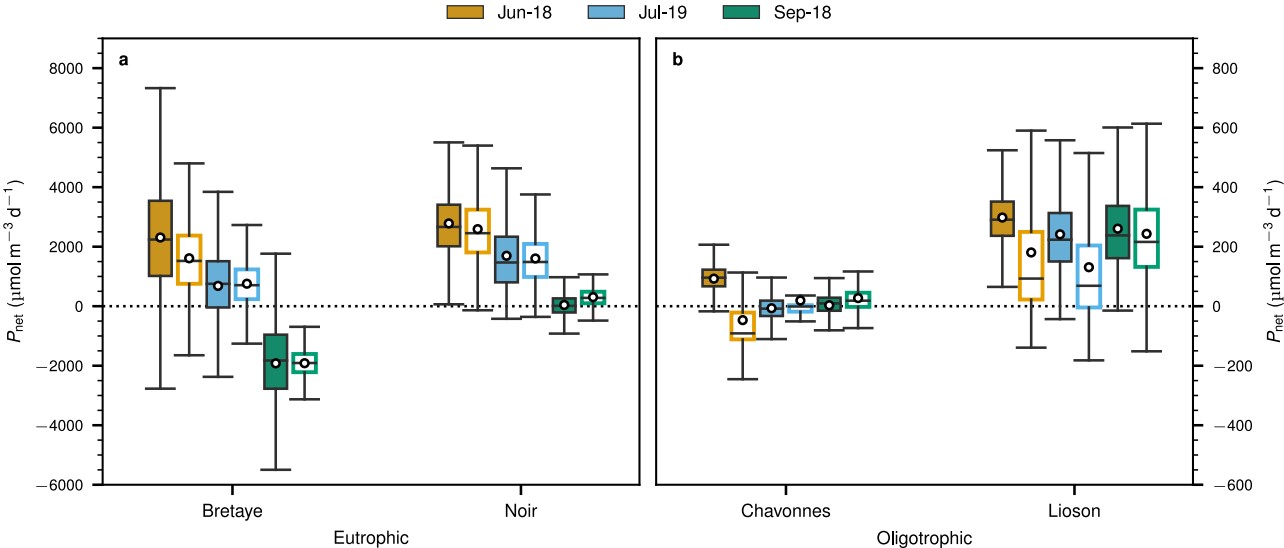

**Fig. 3 | $P_{net}$ rates estimations in the surface mixed layer of each lake using two approaches.** The full-scale mass balance ($P_{net,fs}$; filled boxes) and lateral transport model ($P_{net,lt}$; open boxes). The lakes were divided as **a** eutrophic and **b** oligotrophic lakes. Boxes show the first and third quartiles with the median (line), whiskers extend to most extreme data point within 1.5 times the interquartile range from the box. The white dot represents the average of the $P_{net}$ distribution. Note different scales on y-axes of the two panels.

NOI) were about seven times higher than in the oligotrophic lakes (CHA and LIO). A decrease of $P_{net}$ rates from the beginning to the end of the summer was observed in NOI and BRE, whereas in CHA and LIO $P_{net}$ remained relatively consistent across campaigns.

## Sensitivity analysis of surface diffusive emission to the atmosphere

Several studies have used $k_{600}$ literature parameterizations to estimate $F_a$ (Tan et al.[50] and references therein), although other studies have shown that these estimates often do not correspond with field measurements (Klaus & Vachon[45] and Supplementary Fig. 5). Therefore, we analyzed the impact of $k_{600}$ parameterizations on $P_{net}$ as it is one of the main parameters affecting the mass balance in the epilimnion.

Since the $P_{net}$ results from both models were similar, we used $P_{net}$ from the full-scale mass balance in the following sensitivity analysis. In the lateral transport model (Eq. (2)), we simulated surface $CH_4$ concentrations either with the addition of $P_{net}$ as obtained from the full-scale mass balance approach ($P_{net} = P_{net,fs}$), or without any addition from $P_{net}$ (i.e., $P_{net} = 0$). We also used five different mass transfer coefficient parameterizations ($k_{600}$) to model diffusive $CH_4$ emissions to the atmosphere in the lateral transport model (Table 3). Thus, the resulting surface $CH_4$ concentrations were obtained from the combinations of $P_{net}$ and $k_{600}$, as they determined different boundary conditions of the mass balance in the SML. The analysis is focused on the best and worst fits of the mass transfer coefficient parameterizations (MA10-NB and CC98, respectively) when compared with chamber-based estimations for $CH_4$ ($k_{CH_4}^{cb}$) (Supplementary Figs. 5 and 6). The results of the three remaining parameterization comparisons are available in Table 3 and Supplementary Fig. 10.

The best agreement between measured and simulated $CH_4$ concentrations was found using $P_{net}$ from the full-scale mass balance ($P_{net,fs}$) and $\overline{k}_{CH_4}$ ($P_{net}$-$\overline{k}_{CH_4}$, Table 3, Supplementary Fig. 11b). When using $\overline{k}_{CH_4}$ with $P_{net}$ set to zero ($P_{net}0$-$\overline{k}_{CH_4}$), average $CH_4$ concentrations along the transect were underestimated relative to the measured values (MNB = −1.83, Table 3, Supplementary Fig. 11a). Using $P_{net,fs}$ with the MA10-NB or CC98 parameterizations ($P_{net}$-MA10-NB and $P_{net}$-CC98) resulted in an overestimation of $CH_4$ concentrations (Table 3, Supplementary Figs. 11d, f), whereas when $P_{net}$ was set to zero ($P_{net}0$-

MA10-NB and $P_{net}0$-CC98) with those $k_{600}$ parameterizations, the average $CH_4$ concentrations along the transect were underestimated (Table 3, Supplementary Figs. 11c, e).

## Contribution of methane sources to atmospheric diffusive emissions

The sediment flux ($F_s$) and $P_{net}$ were the two major sources of $CH_4$ in the SML. Using the results obtained from the full-scale mass balance we found that $P_{net}$ contributed ~30% of the $CH_4$ emissions in BRE and CHA, while it reached up to 60% and 90% for NOI and LIO, respectively (Fig. 4). $P_{net}$ was a dominant source in all lakes in June and July except for CHA in July. Negligible $P_{net}$ contributions (<8%) were found in all lakes in September 2019, except for LIO (91%). On average, $F_s$ contributed about 10, 30, 50, and 65% to the $CH_4$ emissions in LIO, NOI, BRE, and CHA, respectively. For CHA and NOI, the $F_s$ contribution increased at the end of the summer and reached up to 90% for CHA in September. For BRE and LIO, the $F_s$ contribution remains relatively constant during the different months. On average, $F_s$ contributed about the same in the oligotrophic and eutrophic lakes. The vertical turbulent flux ($F_z$) contributed about 50% of the atmospheric $CH_4$ emission from BRE and NOI in September and about 30% from CHA in July, but was negligible (<9%) for the other campaigns. The contribution from bubble dissolution ($R_{dis}$) was negligible (<4%) in BRE and NOI and absent in CHA and LIO.

## Discussion

In most of our study lakes, the $P_{net}$ values were positive, indicating that OMP was greater than MOx, and that $P_{net}$ thus acted as a $CH_4$ source during daytime conditions over the stratified season (Fig. 3). $P_{net}$ was near zero in CHA, which is the meso-oligotrophic lake with the largest water level changes throughout the summer, in contrast to the other pre-alpine lakes in our study that maintained relatively consistent water levels. The observed average $P_{net}$ rates were within the range of values previously reported[42], except for NOI with the highest $P_{net}$ rate reported to date (2308 ± 2024 µmol m⁻³ d⁻¹).

$P_{net}$ rates were temporally variable in each lake and varied between study sites. While $P_{net}$ and $\delta^{13}C_{CH_4}$ were relatively constant during the stratified season in the oligotrophic lakes, highly positive $P_{net}$ rates at the beginning of the summer indicated that OMP was an

**Table 3 | Results of the sensitivity analysis of the use of five literature mass transfer coefficients ($k_{CH_4}$), with and without net CH$_4$ production ($P_{net}$), to simulate the CH$_4$ concentrations using the lateral transport model**

| Configuration name | $P_{net}$ | $k_{CH_4}$ parameterizations | RMSE | $R^2$ | MNB |
|---|---|---|---|---|---|
| $P_{net}0\text{-}\overline{k}_{CH_4}$ | 0 | $\overline{k}_{CH_4}$ | 0.81 | 0.65 | −1.83 |
| $P_{net}0\text{-}CC98$ | 0 | $k_{600} = 2.07 + 0.215U_{10}^{1.7}$ [46] | 0.77 | 0.54 | −0.62 |
| $P_{net}0\text{-}MA10\text{-}NB$ | 0 | $k_{600} = 2.045U_{10} + 2$ [20] | 0.78 | 0.59 | −1.56 |
| $P_{net}0\text{-}MA10\text{-}MB$ | 0 | $k_{600} = 2.25U_{10} + 0.16$ [20] | 0.74 | 0.59 | −0.99 |
| $P_{net}0\text{-}MA10\text{-}PB$ | 0 | $k_{600} = 1.75U_{10} - 0.15$ [20] | 0.68 | 0.53 | −0.25 |
| $P_{net}0\text{-}VP13$ | 0 | $k_{600} = 2.51 + 1.48U_{10} + 0.39U_{10}\log_{10}(A_s)$ [47] | 0.85 | 0.61 | −2.11 |
| $P_{net}\text{-}\overline{k}_{CH_4}$ | $P_{net,fs}$ | $\overline{k}_{CH_4}$ | 0.22 | 0.92 | 0.07 |
| $P_{net}\text{-}CC98$ | $P_{net,fs}$ | $k_{600} = 2.07 + 0.215U_{10}^{1.7}$ [46] | 0.57 | 0.82 | 1.72 |
| $P_{net}\text{-}MA10\text{-}NB$ | $P_{net,fs}$ | $k_{600} = 2.04U_{10} + 2$ [20] | 0.39 | 0.79 | 0.64 |
| $P_{net}\text{-}MA10\text{-}MB$ | $P_{net,fs}$ | $k_{600} = 2.25U_{10} + 0.16$ [20] | 0.51 | 0.77 | 1.30 |
| $P_{net}\text{-}MA10\text{-}PB$ | $P_{net,fs}$ | $k_{600} = 1.74U_{10} - 0.15$ [20] | 0.64 | 0.77 | 1.91 |
| $P_{net}\text{-}VP13$ | $P_{net,fs}$ | $k_{600} = 2.51 + 1.48U_{10} + 0.39U_{10}\log_{10}(A_s)$ [47] | 0.35 | 0.79 | 0.03 |

Root mean square error (RMSE), coefficient of determination ($R^2$) and mean normalized bias (MNB) are shown for the comparison between simulated and measured surface CH$_4$ concentration. $P_{net,fs}$ refers to the $P_{net}$ rates obtained from the full-scale mass balance. $k_{CH_4}$ were calculated from the $k_{600}$ literature parameterizations (Eq. (5)) to be used in Eq. (2). $U_{10}$, wind speed at 10 m (m s$^{-1}$); $A_s$, surface lake area (km$^2$); $k_{600}$, gas transfer coefficient (cm h$^{-1}$); $\overline{k}_{CH_4}$ is the average chamber-based mass transfer coefficient. Given the different order of magnitudes of the concentrations measured at each lake, all the statistics were calculated using the logarithm base 10 of each value.

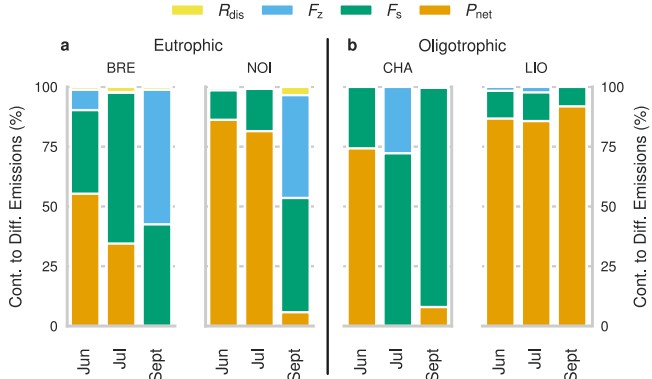

**Fig. 4 | Contribution to diffusive atmospheric CH$_4$ emissions from each component of the CH$_4$ budget.** The sediment flux ($F_s$), diffusive flux from hypolimnion ($F_z$), bubble dissolution ($R_{dis}$), and net production rates ($P_{net}$) in the SML of Lac de Bretaye (BRE), Lac Noir (NOI), Lac des Chavonnes (CHA) and Lac Lioson (LIO). The lakes were divided as **a** eutrophic and **b** oligotrophic lakes. The results from the full-scale mass balance were used as representative $P_{net}$ rates of the studied lakes.

active source of CH$_4$ to the atmosphere in the eutrophic lakes. By the end of the stratified season, $P_{net}$ became negative indicating that MOx was dominating, which was corroborated by isotopically enriched CH$_4$ (Table 1). This seasonal trend in $P_{net}$ was also observed by Günthel et al.[8] and may be related to the CH$_4$ production rates of different algal species[25] and their concentration during the stratified season. In addition, the eutrophic lakes BRE and NOI had $P_{net}$ rates one order magnitude higher than the more oligotrophic lakes (CHA, LIO), suggesting that $P_{net}$ may also be related to trophic state. From this perspective, productive lakes in general may experience higher $P_{net}$ rates than less productive ones.

The dominant sources of CH$_4$ to the surface waters of our lakes were $P_{net}$ and $F_s$, although individual rates of these sources varied across campaigns. Despite eutrophic lakes have generally higher $P_{net}$ rates compared to more oligotrophic ones, the $P_{net}$ contribution fraction to surface diffusive CH$_4$ emissions were independent of the trophic status of the lake. For example, the fraction of $P_{net}$ contribution to emissions was similar and even higher in oligotrophic LIO than that in eutrophic NOI. This was mainly due to the substantial contribution of CH$_4$ from the littoral sediments to the SML in the eutrophic lakes. Therefore, our results suggest that there is no relationship between the

contribution of the two dominant CH$_4$ sources ($P_{net}$ and $F_s$) and trophic state, even though each of these sources are higher in more productive systems.

The methodologies for determining $P_{net}$ are limited by the accuracy of the boundary conditions of the mass balance (i.e., diffusive CH$_4$ emissions at the AWI, CH$_4$ flux from littoral sediment, ebullition, etc.). These boundary conditions are often based on a few measurement locations and are naturally variable. The variability and uncertainty of such estimations led to the observed range of $P_{net}$ in mass balance approaches obtained with the Monte Carlo simulations (Fig. 3). Therefore, to assess the robustness and the validity of the models used, we compared the boundary condition components ($F_a$, $F_s$, and $R_{dis}$) with literature values and examined how their variability may alter the outcome of the two mass balance models.

Diffusive CH$_4$ emissions to the atmosphere are temporally and spatially variable. We accounted for the spatial variability by using the average of ten surface flux measurements along a lake-wide transect for each $P_{net}$ calculation. In addition, the average diffusive CH$_4$ emissions estimated for NOI, CHA, and BRE are well within the range reported for the stratified season of these lakes in previous studies (0.06–4.38 mmol m$^{-2}$ d$^{-1}$; Rinta et al.[31]). There are no previous data for LIO.

A large uncertainty in the estimation of surface diffusive CH$_4$ emissions is due to the parameterization of mass transfer coefficient ($k_{600}$). Therefore, we applied five alternative $k_{600}$ parameterizations to estimate CH$_4$ diffusion at the AWI in the four pre-alpine lakes and compared these fluxes with direct measurements using floating chambers. The comparison of the chamber-based mass transfer coefficient ($k_{600}^{cb}$) with all the tested parameterizations resulted in a low correlation ($R^2 < 0.38$) and clear underestimation of the measured $k_{600}$ values (Supplementary Fig. 5), reflecting the limitations of the $k_{600}$ models across different lakes[45]. The underestimation by $k_{600}$ parameterizations has also been reported in previous studies (Tan et al.[50] and references therein). We hypothesize that the presence of oxygen microbubbles produced by photosynthesis in the water column[51] might enhance the mass transfer coefficient[44]. This phenomenon would be more relevant in high altitudes lakes, such as our study lakes, due to the lower air pressure and oxygen saturation concentration.

In our analysis of the $k_{600}$ parameterizations for the lateral transport model, we observed that when using the literature parameterizations for surface CH$_4$ fluxes, the simulated surface CH$_4$ concentrations were underestimated when $P_{net}$ was not included in the simulations (i.e., $P_{net} = 0$). This is explained by the fact that these

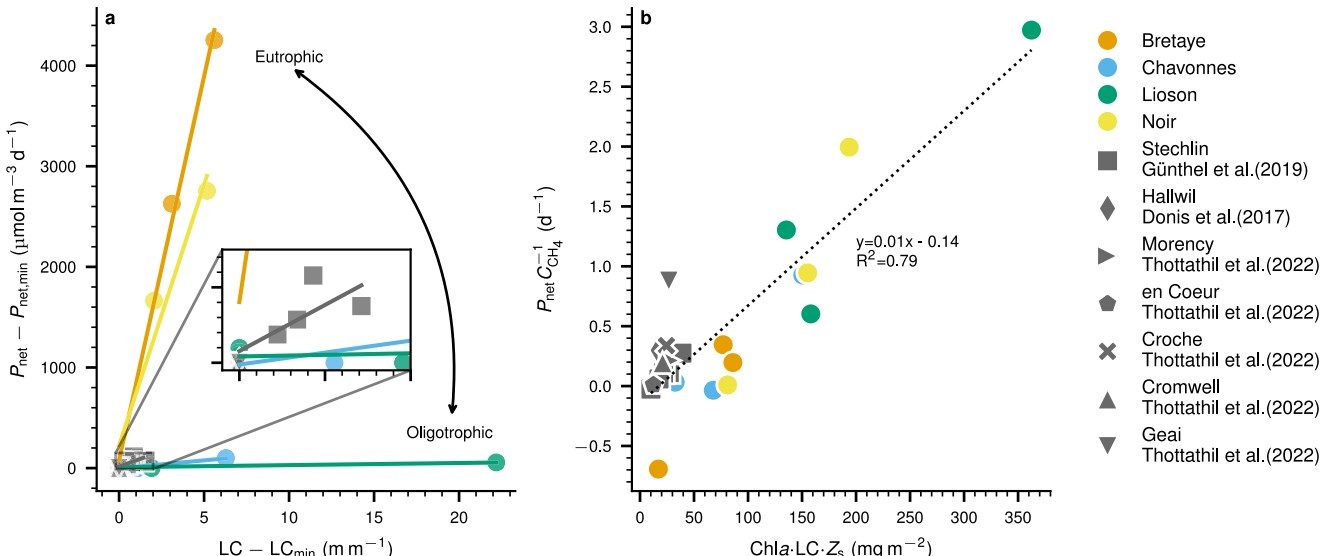

**Fig. 5 | Linking net CH$_4$ production ($P_{net}$) in the surface mixed layer (SML) with trophic variables. a** Relationship between $P_{net}$ and light climate (LC, m m$^{-1}$) and trophic state. Per lake, the minimum $P_{net}$ rate ($P_{net,min}$) and the minimum LC (LC$_{min}$) were subtracted to be able to compare the slope of each curve. $P_{net}$ becomes more independent of LC in more oligotrophic lakes. **b** Interaction between $P_{net}$ (mmol m$^{-3}$ d$^{-1}$) and the average surface concentration of chlorophyll-$a$ (Chl$a$, mg m$^{-3}$), LC (m m$^{-1}$) and Secchi depth ($Z_s$, m) suggest a direct role of photosynthesis on OMP. Specific production/oxidation rate calculated as $P_{net}$ normalized by the average surface concentration of CH$_4$ ($C_{CH_4}$ mmol m$^{-3}$) versus Chl$a$ × light climate ($LC = 2.5 \frac{Z_s}{H_{SML}}$) × $Z_s$; where $H_{SML}$ is the surface mixed layer depth. Chl$a$ was obtained from fluoroprobe profiles measured at the center of the lake. All the parameters were calculated at each sampling campaign. The results from the full-scale mass balance were used as representative $P_{net}$ rates of the studied lakes.

parameterizations underestimate $k_{600}^{cb}$ for all lakes and they do not correlate well with $k_{600}^{cb}$ (Supplementary Fig. 6). In our case, lake-specific (i.e., chamber-based) $k_{600}$ values provided the best results. To further corroborate this finding, we calculated the $P_{net}$ error due to an over- or underestimation of $k_{CH4}$ caused by using $k_{600}$ parameterizations instead of $k_{600}^{cb}$. Our analysis showed a linear relationship between the $P_{net}$ error and the $k_{CH_4}$ error, where the slope is defined by the ratio between the $F_a$ and the $P_{net}$ rates estimated using the measured values (Methods, Supplementary Fig. 12). Excluding the cases when $P_{net}$ was negligible (BRE: Jul-19, CHA: Sept-18 and Jul-19 and NOI: Sept-18), the slope varied between 0.5 to 2.9 with an average value of 1.3, meaning that the $P_{net}$ error is on average 30% higher than the $k_{CH_4}$ error. This result highlights the need to use lake-specific in-situ measured fluxes to compute $k_{600}$ in a mass balance as suggested by various researchers (e.g., Klaus & Vachon[45] and Cole et al.[52]).

The littoral diffusive sediment fluxes were within the range of values reported in the literature (0.001–8.8 mmol m$^{-2}$ d$^{-1}$ [9,37,53]). In the full-scale mass balance, we assumed that the initial lateral flux towards the center of the lakes was equal to the diffusive CH$_4$ flux coming from the littoral sediment (i.e., SML exposed). As the CH$_4$ production rates in sediments increase with increasing temperature[37], it has been hypothesized that sediment CH$_4$ diffusion will also follow this relationship[9]. As most of our sediment flux measurements in the littoral zone were performed in July when the temperatures were highest in all lakes (Supplementary Table 4), we assume that those observed sediment fluxes were on the higher end of possible values. Therefore, using a presumably high sediment flux from July for the mass balance of other months would result in a conservative $P_{net}$ estimate.

Assuming that OMP does not occur (i.e., $P_{net} = 0$) in the full-scale mass balance, the littoral sediment fluxes would have to be two to three times higher than our measured fluxes to compensate for the diffusive CH$_4$ emissions measured at the AWI (Supplementary Fig. 13). In oligotrophic LIO, a littoral sediment flux of about 2.23 ± 1.12 mmol m$^{-2}$ d$^{-1}$ is needed, which is unlikely given that that flux is an order of magnitude higher than what we measured (0.3 ± 0.1 mmol m$^{-2}$ d$^{-1}$). In fact, in BRE we measured one of the highest littoral sediment fluxes yet reported

(8.3 ± 6.7 mmol m$^{-2}$ d$^{-1}$)[9,37,53,54], and we still required one of the highest $P_{net}$ rates ever reported in the literature (June 2018: 2314 ± 2046 µmol m$^{-3}$ d$^{-1}$) to close the CH$_4$ budget. Therefore, littoral sediment CH$_4$ flux alone cannot account for diffusive CH$_4$ emissions in our lakes and OMP needs to be included to close their CH$_4$ budget.

We also conducted a sensitivity analysis on the ebullitive CH$_4$ fluxes ($F_{eb}$, Methods). Assuming that OMP does not occur in the SML, the measured ebullitive fluxes would have to be 42 and 770 times higher for BRE and NOI in June 2018, respectively, to close the mass balance (Supplementary Table 5). These high estimates are due to the low contribution of bubble dissolution given the short contact time between the bubble and the water in the SML, especially within a very shallow SML depth at the beginning of the summer. Hence, even considering $F_{eb}$ two or three times higher than what we estimated, positive $P_{net}$ rates are required to close the SML CH$_4$ mass balance. Moreover, our ebullitive rates are in the same order of magnitude of what has been reported for similar lakes[31,55]. Ultimately, the sensitivity analyses conducted on ebullitive, littoral sediment, and AWI diffusive fluxes suggest that our $P_{net}$ rates are robust, and that OMP is likely a dominant source of atmospheric CH$_4$ from these lakes.

We conducted a first analysis of potential mechanisms behind OMP based on data we collected. Some studies have suggested that Methylphosphonate (MPn) biodegradation could lead to CH$_4$ production in oxic waters of the ocean[1] and lakes[22], specifically in phosphorus-limited environments. In our pre-alpine lakes, however, we did not observe any correlation between $P_{net}$ and phosphorus in the SML (Supplementary Fig. 14a). Another study suggested OMP mechanism is the production of CH$_4$ in nitrogen-limited environments via the transformation of CO$_2$, nitrogen gas, and hydrogen by the nitrogenase enzyme[23] that is commonly present in cyanobacteria. We observed a weak negative correlation between $P_{net}$ and dissolved inorganic nitrogen (DIN) ($R^2 = 0.37$, Supplementary Fig. 14b), which could indicate the use of nitrogen for OMP. However, to our knowledge CH$_4$ production due to nitrogenase activity in cyanobacteria has not yet been observed. Our data do, however, suggest links between OMP and trophic parameters, similar to relationships found in Bogard et al.[5] and Günthel et al.[25].

Considering the importance of the $P_{net}$ contribution to atmospheric $CH_4$ emissions, it is necessary to derive approaches to estimate and upscale $P_{net}$. Günthel et al.[8] proposed that the OMP contribution to diffusive $CH_4$ emissions from lakes can be estimated as a function of littoral sediment area and SML volume. In our study, the $P_{net}$ contribution to diffusive $CH_4$ flux to the atmosphere was highly variable and disagreed with this simple upscaling approach (Supplementary Fig. 15). While it is plausible that the OMP proportion to diffusive emissions may partially depend on lake bathymetry (i.e., the fraction between the sediment area and the SML volume), our results indicate that OMP is a complex phenomenon that is also related to lake trophic properties (e.g., productivity).

We observed that for an individual lake $P_{net}$ can be explained mostly by changes in light climate (LC) (Fig. 5a). LC defines the average light intensity that phytoplankton can be exposed to in the SML during the day[56]. A lower LC means that surface waters are turbid or the lake experiences a deep SML decreasing the average light intensity. In contrast, higher LC implies clearer waters or smaller SML depth, increasing the average light intensities in the SML. We noticed that increases in LC strongly increase $P_{net}$ rates in eutrophic lakes whereas in oligotrophic lakes $P_{net}$ is nearly independent of LC (Fig. 5a). Recent evidence indicates that OMP could be a photosynthesis-derived process[6,24,25]. Therefore, we hypothesize that the $P_{net}$-LC relationship could also indicate the inhibition of MOx at high-light intensities[12,40] and/or enhanced $CH_4$ production due to production of reactive oxygen species by photoautotrophs at high-light intensities[57].

The $P_{net}$ versus LC relationship strongly depends on the trophic state of each lake and thus cannot alone be used to upscale $P_{net}$ in different lake ecosystems. We suggest an empirical approach using additional trophic state parameters (Fig. 5b). $CH_4$ concentrations (and often $CH_4$ emissions) are dependent on trophic state, as indicated by higher $CH_4$ concentrations typically observed in eutrophic lakes relative to oligotrophic lakes[27]. Therefore $CH_4$ concentration in the SML can be used as a proxy to reflect the trophic state of each lake and to normalize $P_{net}$ rates found in the eutrophic and oligotrophic lakes (Fig. 3). This interaction between $P_{net}$ normalized by the SML $CH_4$ concentration versus $Chla \times LC \times$ Secchi depth indicates the direct role of phytoplankton and light availability in OMP[6,24,25]. Including the data from Donis et al.[7], Günthel et al.[8], and Thottathil et al.[58], this parameterization explains around 80% of the dataset ($R^2 = 0.79$, Fig. 5b). While more data are needed, this provides an important step towards estimating $P_{net}$ in the SML that helps to define OMP dynamics across systems, identify lakes with potentially high OMP rates, and work towards a global upscaling of OMP (or $P_{net}$).

In this study, we quantified the $P_{net}$ rates of $CH_4$ (i.e., net balance between OMP and MOx) in the oxic SML of four pre-alpine lakes using two models that have previously produced contradictory results when resolving OMP in lowland lakes[7–9,41,42]. The good agreement between the adaptation of these approaches used in our study shows that there are no methodological issues with the models themselves when the appropriate boundary conditions are used to estimate OMP (or $P_{net}$, in our case). We also conducted a thorough sensitivity analysis on the three main parameters that lead to the highest uncertainties. This analysis shows that measured surface fluxes must be used instead of literature $k_{600}$ parameterizations to estimate the diffusive $CH_4$ flux to the atmosphere. Our results indicate that in three out of four lakes a positive $P_{net}$ (i.e., a net input of $CH_4$ from OMP) needs to be included in the SML $CH_4$ budget. In fact, up to 85% of atmospheric $CH_4$ emissions that occurred at the beginning of summer resulted from $P_{net}$, and even in systems with some of the highest recorded littoral sediment fluxes, we still obtained some of the highest reported $P_{net}$ (or OMP) rates.

Finally, while the mechanisms behind OMP need further investigation, this study (in agreement with previous ones[6,12,24,25]) show that light and photoautotrophs may play a significant role in OMP.

Consequently, future changes in light availability and temperature may induce positive feedbacks by promoting algal species capable of producing $CH_4$. Although the contribution of OMP to total diffusive emissions from inland waters is still not well constrained, we have shown that it can be a dominant source from lakes in the pre-alpine region where climatic changes occur at higher rates than the global average[33,34]. It is thus crucial to continue quantifying the contribution of $P_{net}$ from various aquatic systems and identifying the main drivers of OMP that will help to better understand the impact of OMP on the global $CH_4$ cycle and how to predict or possibly mitigate its impact in a changing climate.

## Methods

### Study sites

Lac de Bretaye (BRE), Lac Noir (NOI), Lac des Chavonnes (CHA), and Lac Lioson (LIO) are pre-alpine lakes (above 1600 m.a.s.l) located in Canton Vaud, Switzerland (Supplementary Table 1). All lakes are of glacial origin and have a wide-range of trophic states (oligotrophic-eutrophic). BRE, NOI, and CHA are ~500 m away from each other, while LIO is located ~7 km away from the others. BRE and NOI are small and shallow lakes without inflow or outflow streams located in alpine meadows used for animal grazing. CHA has a small inflow stream while LIO has a small creek outflow that is the origin of the Hongrin River.

### Limnological measurements

During each campaign, water column profiles were measured at the deepest point of each lake (M1, Supplementary Fig. 1) with a CTD profiler (Conductivity-Temperature-Depth, Seabird SBE19plus) equipped with temperature, conductivity, oxygen, PAR, turbidity, Chl$a$, and pH sensors, and a spectrofluorometer (bbe Moldaenke GmbH, Schwentinental, Germany) to measure total Chl$a$ concentrations.

Total (TP) and dissolved phosphorus (DP), dissolved inorganic nitrogen as nitrate plus nitrite (DIN), dissolved silica (DSIL), and total carbon concentration (TC) were measured at each campaign in the upper mixed layer (from the surface to the bottom of the thermocline) and in the hypolimnion (Supplementary Table 6). Water samples were collected with a Niskin sampler and equal amounts of water from several depths were transferred into two 1 L glass bottle (Duran, GmbH, Mainz, Germany). 50 mL of water was filtered through 0.45 μm (PES) syringe filters to measure dissolved nutrient fractions. An AQ2 Discrete Analyzer (SEAL Analytical) based on spectophotometric methods was used to measure TP and DP by Acidic molybdate/antimony with ascorbic acid reduction[59], Nitrate-N plus Nitrite-N by Cadmium coil reduction followed by sulfanilamide reaction in the presence of N-(1-naphthylethylenediamine)[59] and DSIL by Acidic molybdate with ANSA reduction[60]. A Shimadzu carbon analyzer (TOC-L$_{CPH/CPN}$) measured TC.

### Mass balance

$P_{net}$ in the SML was estimated using two independent mass balance approaches: a 0-D full-scale mass balance following Donis et al.[7] and a 1-D lateral transport model adapted from Peeters et al.[9].

**Full-scale mass balance.** The full-scale mass balance approach assumes that at each sampling date the surface layer can be modeled as a well-mixed reactor and $P_{net,fs}$ can be estimated as follows:

$$\frac{\partial C}{\partial t} \forall_{SML} = A_s F_s - A_a F_a + A_z F_z + R_{dis} \forall_{SML} + P_{net,fs} \forall_{SML}; \quad [mol\ d^{-1}] \quad (1)$$

where $C$ is surface $CH_4$ concentration, $\forall_{SML}$ is SML volume, and $A_s$, $A_a$, and $A_z$ are sediment area, lake surface area, and planar area at the bottom of the SML (Supplementary Table 7), respectively. The spatial average values for the surface fluxes ($F_a$), bubble dissolution rates ($R_{dis}$) in the SML, and hypolimnetic fluxes ($F_z$) were used as boundaries conditions (Table 2). A sonar survey was performed to obtain the bathymetry of each lake (Supplementary Fig. 1) and $A_a$, $A_s$, and $A_z$ were

determined using the software Surfer® (Golden Software, LCC) (Supplementary Table 7). The bottom of the SML ($H_{SML}$) was defined when $\partial T/\partial z$ becomes smaller than $-1\,°C\,m^{-1}$ [61] (Table 1). The net $CH_4$ production ($P_{net}$) in the SML was estimated using Eq. (1) assuming steady-state conditions ($\frac{\partial C}{\partial t}\forall_{SML} = 0$) and that the lateral contribution to the mass balance is equal to the littoral sediment flux times the area of the sediment.

**Lateral transport model.** Using a modified version of the lateral transport model presented by Peeters et al.[9], $P_{net,lt}$ rates for each lake were obtained by finding the simulated transect $CH_4$ concentrations that best-fit to the measured $CH_4$ concentrations. In this study, the lateral transport model includes vertical diffusive $CH_4$ flux through the bottom of the SML and bubble dissolution:

$$\frac{\partial C(r)}{\partial t} = K_H \frac{1}{H(r)r}\frac{\partial}{\partial r}\left(H(r)r\frac{\partial C(r)}{\partial r}\right) + \frac{1}{H(r)}K_z\frac{C_{hyp} - C(r)}{\Delta z} - \frac{\bar{k}_{CH_4}}{H(r)}$$
$$\left(C(r) - H_{cp}pCH_{4,atm}\right) + \frac{F_s(r)}{H(r)} + R_{dis}(r) + P_{net,lt}; \quad [mol\,m^{-3}\,d^{-1}] \quad (2)$$

where $H(r)$ is the spatially varying thickness of the SML. The mass transfer coefficient for $CH_4$ was calculated based on the average gas transfer coefficient obtained from the flux chambers ($\bar{k}_{CH_4}$), $C_{hyp}$ is the $CH_4$ concentration 1 m below the bottom of the SML, $\Delta z = 1\,m$, $pCH_{4,atm}$ is the partial pressure of atmospheric $CH_4$ and $H_{cp}$ is the Henry constant of $CH_4$ dissolution at in-situ temperature. This model considers that the surface layer is fully mixed in the vertical and, therefore, the vertical $CH_4$ concentrations are homogeneous within the SML.

In the simulations of each lake, we assumed that the SML, sources, and sinks are radially symmetric in the horizontal plane. Therefore, the development of $CH_4$ concentration can be described based on the radial distance $r$ from the shore to the center of the lake ($r_{max} = \sqrt{A_a/\pi}$).

Two regions were defined in the model, the littoral zone ($r \leq r_s = \sqrt{(A_a - A_s)/\pi}$) and the pelagic waters ($r > r_s$). The SML thickness ($H(r)$) is equal to the mixed layer depth in the pelagic region and, within the littoral zone, $H(r)$ decreases linearly with $r$ from the mixed layer depth to zero at the shore. The littoral sediment flux is zero in the pelagic zone ($r < r_s$) and equal to the measured average littoral sediment flux ($\bar{F}_s$) in the shallow region ($r \geq r_s$) as:

$$F_s(r) = \begin{cases} \bar{F}_s & \text{for } r \geq r_s \\ 0 & \text{for } r < r_s \end{cases} \quad [mmol\,m^{-2}\,d^{-1}] \quad (3)$$

Average bubble dissolution rates ($R_{dis}(h(r))$) as a function of lake depth ($h$) were included in the SML. At the boundaries, horizontal fluxes were assumed as zero. To estimate the horizontal dispersion coefficient ($K_H$) we used Peeters & Hofmann[49] parameterization:

$$K_H = 1.4 \times 10^{-4}L^{1.07} \quad [m^2s^{-1}] \quad (4)$$

where the length scale $L$ [m] was calculated as $L = r_s$ (Supplementary Table 7). Eq. (4) is the average of the results 1, 3, and 4 found in Table 2 of Peeters & Hofmann[49].

$P_{net}$ rates were obtained using least square method optimization solver implemented with the *curve fit* function from Scipy[62] in Python.

**Monte Carlo simulation**
To assess uncertainties, Monte Carlo simulations were performed (10,000 iterations) when solving the full-scale mass balance and the lateral transport models. $P_{net}$, $R_{dis}$, and $F_z$ were selected within a normal distribution resulting from the mean ($\mu$) and their standard deviation (SD) retrieved from the field measurements. Given the small contribution of $R_{dis}$ to the $CH_4$ in the SML, its variability was not included in the Monte Carlo simulations of the lateral transport model. To prevent

negative values, $F_a$ and $F_s$ were chosen from a gamma distribution defined by shape ($\kappa = \mu^2/SD^2$) and the scale ($\theta^2 = SD^2/\mu$). Here the gamma distribution has the density $f(x) = (x^{\kappa-1}\frac{e^{-x/\theta}}{\theta^\kappa\Gamma})$ where $\Gamma$ is the gamma function. Random.normal and random.gamma functions from the Numpy package[63] in Python were used for each normal and gamma distributions, respectively.

**Water column $CH_4$ and $\delta^{13}C_{CH_4}$ signature**
At each sampling campaign $CH_4$ and $\delta^{13}C_{CH_4}$ concentration profiles were taken at the deepest location of each lake (M1, Supplementary Fig. 1) and along a transect composed of 10–11 stations across the lake (shore to shore, T1–T11, Supplementary Fig. 1).

Dissolved $CH_4$ concentration profiles were performed at a maximum depth resolution of 0.5 m where the metalimnetic $CH_4$ gradient was expected. For the profile, the water samples were obtained with a 5-L Niskin bottle and then gently transferred into a 1-L glass bottle (Duran GmbH, Mainz, Germany) while for the transect the samples were obtained directly with a 1-L glass bottle (Duran GmbH, Mainz, Germany). For both methodologies, the water was overflowing to replace the volume three times. $CH_4$ concentrations and $\delta^{13}C_{CH_4}$ were measured using the headspace method[7]. The samples were measured on a Cavity Ring-Down Spectrometer analyzer (Picarro G2201-i, Santa Clara, CA, USA) for $CH_4$ concentrations in the gas phase (ppm) and stable isotope ratio ($\delta^{13}C_{CH_4}$ in ‰). Water $CH_4$ concentrations were back-calculated according to Wiesenburg & Guinasso[64] accounting for water temperature, air concentration, and the headspace/water ratio (500 mL air/500 mL water) in the bottle.

**$CH_4$ diffusive fluxes to the atmosphere**
Diffusive $CH_4$ emissions to the atmosphere ($F_a$) were measured using a floating chamber attached to a portable GHG analyzer (UGGA; Los Gatos Research, Inc.). Instrument-specific precision at ambient concentrations ($1-\sigma$ of 100 s average) for [$^{12}CH_4$] is 0.25 ppb. The floating chamber consists of an inverted plastic container with foam elements for floatation (as in McGinnis et al.[44]). To minimize artificial turbulence effects, the buoyancy element was adjusted that only ~2 cm of the chamber penetrated below the water level. The chamber was painted white to minimize heating. Two gas ports (inflow and outflow) were installed at the top of the chamber via two 5 m gas-impermeable tubes (Tygon 2375) and connected to the GHG analyzer measuring the gaseous $CH_4$ concentrations in the chamber every 1 s. Transects were performed with the chamber deployed from a boat. The chamber was allowed to freely drift to minimize artificial disturbance. Fluxes were obtained by the slopes of the resolved $CH_4$ curves over the first ~5 min when the slopes were approximately linear ($R^2 > 0.97$).

To simulate the fluxes to the atmosphere in the lateral transport model, chamber-based mass transfer coefficient ($k_{CH_4}^{cb}$) was estimated using the chamber-based surface fluxes and Fick's 1st Law[44] as:

$$F_a = k_{CH_4}\left(C_w - H_{cp}pCH_{4,atm}\right); \quad [mmol\,m^{-2}\,d^{-1}]$$
$$k_{CH_4} = k_{600}(600/Sc)^n; \quad [m\,d^{-1}] \quad (5)$$

where $C_w$ is the $CH_4$ concentration in the surface water, Sc is the Schmidt number for $CH_4$ and the exponent is taken as $n = 2/3$ for wind speed $< 3.7\,m\,s^{-1}$ and $n = 1/2$ for wind speed $> 3.7\,m\,s^{-1}$ [44].

**Sensitivity analysis of $k_{CH_4}$ on $P_{net}$ estimation**
We calculated the error on $P_{net}^{err}$ caused by an inaccuracy on the estimation of $k_{CH_4}$ due to the use of $k_{600}$ literature parameterization as:

$$P_{net}^{err} = \frac{P_{net} - P'_{net}}{P_{net}}; \quad [-] \quad (6)$$

where $P'_{net}$ is calculated using Eq. (1) considering $F'_a = k'_{CH_4}(C_w - C_{sat})$, then:

$$P_{net}^{err} = \frac{F_a A_s}{P_{net} \forall_{SML}} k_{err}; \quad [-] \tag{7}$$

where $F_a$ is the average measure flux to the atmosphere and $k_{err} = \frac{k_{CH_4}^{cb} - k'_{CH_4}}{k_{CH_4}^{cb}}$ is the error between the mass transfer coefficient obtained from $k_{600}$ parameterization ($k'_{CH_4}$) and from chamber measurements ($k_{CH_4}^{cb}$).

## Porewater $CH_4$ concentration and $\delta^{13}C_{CH_4}$ signature

Littoral sediment cores were taken in most of the lakes, except for LIO where the rocky bottom made it impossible to take a sample. Sampling was performed with a gravity sediment corer (Uwitech, Mondsee, Austria) equipped with an acrylic liner of 70 cm in length and with an internal diameter of 6 cm. 3 mL of sediment was sub-sampled at 1–2 cm depth intervals with headless 3 mL syringes through the pre-drilled holes from the selected depths. The sediment sub-sample was immediately placed into 1 L glass bottle (Duran GmbH, Mainz, Germany) containing 500 mL of lake water previously bubbled with air to reach equilibrium with the atmosphere. The subsequent procedure followed the same as for the water column headspace method. Porewater $CH_4$ concentrations were back-calculated from the headspace concentrations accounting for dilution of sediment porewater in the lake water (assuming that aerated lake water is in equilibrium with the atmosphere), temperature, headspace ratio, and assuming a porosity of 0.9. The location and depth of each core are shown in Supplementary Fig. 1 and Supplementary Table 4.

## Methane benthic fluxes

The littoral $CH_4$ sediment flux ($F_s$) at each lake was determined as the average flux provided by two independent methods. On average, three cores above the thermocline depth were taken in the epilimnion on September 2018 and July 2019 (Supplementary Table 4) to estimate the littoral sediment fluxes at each lake.

**Porewater method.** Methane fluxes at the sediment-water interface were calculated using the $CH_4$ concentration retrieved from porewater cores and Fick's 1st Law over the linear top 2–3 cm of the porewater concentration profile.

$$F_s = -\phi D_{CH_4} \theta^{-2} \frac{\partial C}{\partial z}; \quad [mmol\, m^{-2}\, d^{-1}] \tag{8}$$

where $F_s$ is the diffusive $CH_4$ flux at the sediment-water interface, $\phi$ the porosity of the sediments (assumed as 0.9), $D_{CH_4}$ the diffusion coefficient for $CH_4$ in water ($1.5 \times 10^{-5}$ cm$^2$ s$^{-1}$ [65]), $\theta^2$ the square of tortuosity (1.2)[66] and $\partial C / \partial z$ the measured vertical concentration gradient.

**Benthic chamber.** Benthic fluxes were measured directly in sediment cores retrieved from the littoral sediment or core liners deployed in situ connected to a portable GHG analyzer (UGGA: Los Gatos Research, Inc.). The core was covered leaving ~5 cm of headspace and ~30–50 cm of water. The lid was connected to a GHG analyzer creating a closed loop and partial pressure of $CH_4$ ($P_{CH_4}$) in the headspace was measured over time. Water $CH_4$ concentrations ($C_w$) were measured at the beginning and at the end of the deployment. Each deployment lasted about 1 h while the surface water was gently stirred to increase the mass transfer coefficient ($k_{bc}$) at the air-water interface without producing sediment resuspension. The sediment flux was calculated using three methods:

- Integrated mass balance: $F_s$ is obtained using the beginning and final air and gas $CH_4$ concentration and performing a mass balance in the water and the air phase as:

$$F_s A_{bc} = \frac{V_{air}}{RT_a} \frac{\Delta P_{CH_4}}{\Delta t} + \frac{V_w \Delta C_w}{\Delta t}; \quad [mmol\, d^{-1}] \tag{9}$$

where $V_w$ and $V_{air}$ are the volume of the water and air phases, respectively. $R$ is the ideal gas constant, $T_a$ is the air temperature and $A_{bc}$ is the surface area of the chamber.

- Transient mass balance: solving the mass balance over time we obtain that:

$$\frac{\partial P_{CH_4}}{\partial t} = \frac{aRT_a}{b}\left(wF_s - (wF_s - bk_{bc}C_o)e^{-bk_{bc}t}\right); \quad [Pa\, d^{-1}] \tag{10}$$

where $w = A_{bc}/V_w$, $a = A_{bc}/V_a$, $C_0 = C_w(0) - H_{cp}P_{CH_4}$ and $b = (w - H_{cp}RT_a a)$. The sediment flux is estimated fitting $k_{cb}$ and $F_s$ to the measured $\partial P_{CH_4}/\partial t$ using least square method optimization solver implemented on the *curve fit* function from Scipy[62] in Python. The $k_{bc}$ boundaries were set from 0–40 m d$^{-1}$ for the fitting.

- Equilibrium mass balance: after ~1 h of measurements, we assume that the exponential part of the curve of Eq. (10) becomes negligible. Therefore, $F_s$ can be estimated with the last 5 min of the $CH_4$ partial pressure as:

$$P_{CH_4} = \frac{aRT_a}{b} wF_s t; \quad [Pa] \tag{11}$$

The flux from the benthic chamber was calculated as the average of the results of the three methods described above.

## $CH_4$ bubble dissolution and ebullition rates

The $CH_4$ dissolution from a single bubble released from the sediment was calculated using McGinnis et al.[48]. For each bubble we considered a diameter of 5 mm and the water column $CH_4$, $CO_2$, and $O_2$ concentrations and temperature profiles. The initial bubble composition at each depth was estimated from a linear interpolation from bubble content obtained following the same methodology as Langenegger et al.[19]. The total bubble dissolution rate ($R_{dis}(z)$) was calculated considering the contribution from all bubbles released below that depth as:

$$R_{dis}(z) = \frac{\sum_{bottom}^{z} r_i \frac{F_{eb,SWI,i}}{n_{0,i}} \Delta A_{sed,i}}{A_p(z)} \quad [\mu mol\, m^{-3}\, d^{-1}] \tag{12}$$

where $r_i$ is the bubble dissolution from an individual bubble at depth $i$ ($\mu$mol bub$^{-1}$), $F_{eb,SWI,i}$ is the $CH_4$ ebullition flux released at the sediment-water interface (SWI) at depth $i$ (mmol m$^{-2}$ d$^{-1}$) and $n_{0,i}$ is the initial amount of $CH_4$ in a single bubble ($\mu$mol bub$^{-1}$). $\Delta A_{sed,i}$ is the sediment area between the depth interval $i$ to $i+1$ (m$^2$). $F_{eb,SWI,i}$ was estimated using Langenegger et al.[19]'s model. Using a mass balance in the sediment, this model predicts $CH_4$ ebullition if the following are known: (1) the bubble $CH_4$ content, (2) the water depth where the bubble was collected and (3) the diffusive $CH_4$ flux from the sediment. In our study, we used the measured $F_s$ to estimate $F_{eb,SWI}$ using Langenegger et al.[19] approach. The mass balance model can be described by:

$$\phi D_i \frac{\partial^2 C_i(z)}{\partial z^2} + W_i(z) = 0, \quad 0 < z < z_{eb,min} \tag{13}$$

$$\phi D_i \frac{\partial^2 C_i(z)}{\partial z^2} + W_i(z) - E(z)\frac{K_{H,i}C_i(z)}{P} = 0, \quad z > z_{eb,min} \tag{14}$$

where $W_i(z)$ (mol m$^{-3}$ d$^{-1}$) is the gas production rates as a function of the sediment depth (assumed exponential for CH$_4$ and zero for the other gases), $z_{eb,min}$ the depth of a nonebullitive layer at the top of the sediment. $D_i$ the molecular diffusion corrected by tortuosity, $C(z)$ is the dissolved concentration (mol m$^{-3}$), $E(z)$ the total gas ebullition per bulk volume (mol m$^{-3}$ d$^{-1}$), $K_{H,i}$ is Henry's law volatility constant (Pa m$^{-3}$ mol$^{-1}$), and $P$ is the local critical gas pressure (Pa).

### Sensitivity analysis of ebullition

We calculated the CH$_4$ ebullition fluxes needed ($F_{eb,need}$) to compensate the $P_{net}$ rates. We selected $P_{net}$ rates for BRE and NOI for June 2018, where we estimated the percentage of $F_{eb}$ that is dissolved in the SML ($\beta$) using McGinnis et al.[48]'s model assuming a bubble diameter of 5 mm. Then $F_{eb,need}$ was estimated using Eq. (15) and the results are summarized in Supplementary Table 5.

$$F_{eb,need} = \frac{P_{net} V_{SML}}{\beta A_{sed}}; \quad [\text{mmol m}^{-2} \text{d}^{-1}] \quad (15)$$

### Vertical diffusive CH$_4$ flux from/to hypolimnion

To estimate the transport of CH$_4$ into the SML via turbulent diffusion we applied Fick's First Law as:

$$F_z = -K_z \frac{\partial C}{\partial z}; \quad [\text{mmol m}^{-2} \text{d}^{-1}] \quad (16)$$

where $F_z$ is the average vertical CH$_4$ diffusive flux, $z$ is depth (m), $\frac{\partial C}{\partial z}$ is the vertical gradient measured at 1 m depth resolution approximately. The vertical diffusivity ($K_z$) was determined at each lake for each campaign (Supplementary Fig. 16) from temperature CTD profiles (sampling rate 4 Hz) and the Osmidov method[67] as:

$$K_z = \gamma_{mix} L_T^2 N; \quad [\text{m}^{-2} \text{d}^{-1}] \quad (17)$$

where $\gamma_{mix}$ is the mixing efficiency (assumed 0.15, Wüest & Lorke[68]), $N$ is the Brunt-Väisälä buoyancy frequency and $L_T$ is the Thorpe scale estimated from the maximum displacement length ($L_{max}$) as Lorke & Wüest[69]:

$$L_T = \frac{\sqrt{2}}{7.3} L_{max}; \quad [\text{m}] \quad (18)$$

This estimation was tested using microstructure profiles measured with a self-contained autonomous microstructure profiler (SCAMP; PME, Inc.) during the summer of 2021 in BRE, NOI, and CHA (Supplementary Fig. 17), where turbulence profiles were resolved after Kreling et al.[70].

### Contribution to total diffusive CH$_4$ emissions

We studied the importance of each source contribution (SC) to the diffusive surface flux by computing:

$$SC_i = \frac{S_i}{\sum_j S_j} \cdot 100; \quad [\%] \quad (19)$$

where $S_i$ is each source term (mol d$^{-1}$) such as bubble dissolution ($R_{dis} \forall_{SML}$), sediment flux ($F_s A_s$), net production ($P_{net} \forall_{SML}$), and vertical diffusive fluxes ($F_z A_z$). If $S_i \leq 0$ then $S_i = 0$ where $i$ is each source term.

## Data availability

All relevant data included in this manuscript are available in https://doi.org/10.5281/zenodo.7691859.

## Code availability

The code for the lateral transport model can be found in https://doi.org/10.5281/zenodo.7695166.

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

## Acknowledgements

We would like to thank Alexandrine Massot, Aurora Pinto, Kam Tang, Roxane Fillion, and Sabine Flury for their help during the fieldwork and laboratory measurements, and the Canton of Vaud, Direction générale de l'environnement (DGE) and the Municipalité d'Ormont-Dessous for providing access to all sampled lakes. We would like to thank Marco Günthel and Shoji Thottathil for providing us with data for the upscaling approach. Funding for this study was provided by the Swiss National Science Foundation (SNSF) to D.F.M. Grants No. 200021_169899 (Methane Paradox)- and European Union's Horizon 2020 research and innovation program under the Marie Skłodowska-Curie grant agreement to T.D. No 788612 (TRIAGE).

## Author contributions

C.O., T.D., and D.F.M. initiated and designed the study, organized campaigns, performed sampling, and analyzed the data with significant contributions from T.L., D.D., and E.S. C.O. wrote the manuscript with editorial help and conceptual contribution from all the co-authors.

## Competing interests

The authors declare no competing interests.
