## [Peer Review File · Nature Communications]

Evaluation of the methane paradox in four adjacent pre-alpine lakes across a trophic gradientReviewer #1 (Remarks to the Author):

MAJOR COMMENTS

The authors present in an attractive way CH₄ data reported in 4 alpine lakes. They compute to close a mass balance what they call net CH₄ production (P_{net}) corresponding to oxic methane production (OMP) minus methane oxidation (MOX). They then use their data with another two studies to derive a general relation between P_{net} and chlorophyll-a concentration (plus light regime indicated by Secchi depth). This relation is supposed to contribute "towards a global scaling of oxic methane production" as claimed in the title.

There are several problems in the approach.

First, P_{net} is computed as an input term to close the mass balance of several input/output terms. This means that if an output term is over-estimated, mathematically P_{net} is automatically over-estimated. This seems to be the case for the diffusive emission of CH₄ to the atmosphere derived from floating chambers that are known to provide over-estimates of gas emissions (This is quite intuitive if you picture mentally a chamber being thrown around by waves or dragged by wind). Alternatively if another input term is under-estimated, mathematically P_{net} is also automatically over-estimated. This seems to be the case of CH₄ input in the mixed layer from dissolution of CH₄ from rising bubbles (originating from sedimentary ebullition). CH₄ ebullition was not actually measured but seems to have been computed from sediment data in a way that is briefly mentioned in text in an obscure way. So it is possible (and in my opinion very likely) that the reported P_{net} are vastly over-estimated.

Second, the authors attempt to derive a "general" relation between P_{net} and Chlorophyll and light regime. The problem is that P_{net} is normalized by CH₄. On the one end, there is no conceptual reason for such a normalization, in fact it does not make sense at all to do such a normalization. On the other hand it's the normalization itself that most probably drives the relationship, and that non-normalized P_{net} is unrelated to Chlorophyll and light regime.

Third, to derive a "general" relation between P_{net} and Chlorophyll, the authors use the data of Guenther et al. (2019) and Donis et al. (2017) that were shown recently by Peeters and Hoffman (2021) to actually be erroneously over-estimated. I find it puzzling that authors used discredited data in their analysis.

Fourth, to derive a "general" relation between P_{net} and Chlorophyll, the authors have excluded the independent study of Morana et al. (ref 46) that reported data that could have been used in attempting to build a general relation between P_{net} and Chlorophyll and light regime.

137-141 : I find it very worrying that chamber estimates of K₆₀₀ are over-estimated compared to 5 independent parameterizations. For me this is indicative that the measurements reported by the authors are over-estimated due to a methodological bias. And indeed, chamber measurements are well known to over-estimate fluxes due to pressure and temperature changes in the chamber during the deployment (Belanger and Korzum 1991) or to artificial enhancement of water turbulence due to the movement of the chamber (Vachon et al. 2010). The movement of the chamber can be due to the oscillation of the chamber with waves or the chamber being dragged by the wind. What is also worrying is that the authors do not present any convincing physical explanation for the fact that these 4 lakes would be so special as to have higher K₆₀₀ values. For instance the Cole and Caraco parameterization is based on the compilation of numerous tracer experiments in numerous lakes of variable surface area and depth. So this parameterization should cover the typical range of physical attributes of lakes.

This issue is not marginal, since net OMP is calculated (as an input) by difference in a mass balance in which the diffusive emission to the atmosphere is an output term. If one the output term is erroneously over-estimated, automatically net OMP is also erroneously over-estimated since it is computed to close the mass balance.

In a similar note, an important input of CH₄ to the oxygenated mixed layer is the dissolution of CH₄ from the rising bubbles from the sediment. This is an important process because it allows CH₄ produced in the sediments to enter the oxygenated mixed layer, bypassing water column methane oxygenation. This process alone might in theory explain all of the reported OMP that is calculated to close a mass balance. For instance the reported higher OMP in the two eutrophic lakes (BER and NOI) would be consistent with a higher ebullition in these lakes (higher sediment organic matter) and dissolution of CH₄ in the oxygenated mixed layer from rising bubbles. The problem is that it is unclear from the ms how the ebullitive fluxes were actually derived. From text in line 156, it seems that ebullitive fluxes were somehow modelled from diffusive fluxes. It would be useful if the authors could explain in much more detail how exactly this was achieved, and what would be the error associated to these ebullitive fluxes. Similarly it would be useful if they provide an error propagation estimate on OMP (as done for the air-water estimates).

Available studies show that OMP is linked to phytoplankton metabolism (papers from Grossart, Bizic, etc). The exact nature of this link is not fully resolved and as rightly pointed out by the authors it could be multiple and variable from one lake to another. Yet, whatever the mechanism there should be a link between phytoplankton biomass and OMP. This was shown in the (selection of) vertical profiles shown in the Grossart et al. (2011) paper, showing a relation between the peaks of CH₄, Chlorophyll-a and O₂ at the thermocline in lake Stechlin. I'm surprised that the chlorophyll-a profiles from the lakes are not shown in Figure S2. On the contrary to what would be expected, the lack of relation between O₂ and CH₄ suggests there are no functional relations as would be expected from OMP and primary production.

On the contrary, the vertical structure of the concentration of CH₄ and of the stable isotope of CH₄ show "classical" patterns that can be interpreted as resulting from the inputs of CH₄ from sediments (diffusion or dissolution of bubbles).

L306: I would like to see the graph of CH₄ concentration in the SML versus Chla X LC x Secchi depth. If CH₄ correlates given that CH₄ typically changes over 1 or 2 orders of magnitude in the SML, the relation between Pnet and Chla X LC x Secchi depth could simply result from the normalization.

Also, from a conceptual point of view this normalization is very strange. Trophic status correlates with Chla (and light). Since Pnet correlates to trophic status (as stated), then non-normalized Pnet should correlate to Chla X LC x Secchi. So why did you normalize Pnet for trophic status? This simply does not make sense from a conceptual point of view. The only use of such an approach would be if CH₄ strongly relates to Chla X LC x Secchi, and the normalization allows to artificially derive a relationship between normalized Pnet and Chla X LC x Secchi.

L 309: Statement "While more data are needed to understand this parameterization" is intriguing. Morana et al. (2021) (ref 46) reported OMP and MOX (allowing to compute Pnet) as well as Chla and Secchi depth. This was reported in 3 lakes. This would allow to substantially increase the data-set given the actually low number of data points used by the authors to derive their relation. Also the work of Morana spans a large gradient of productivity and in a tropical climate that would allow to really contribute "towards a global scaling".

MINOR COMMENTS

L25-26 : « methane paradox » concept has only been recently extended to lakes. Over-saturation of CH₄ in surface waters of lakes (that are oxic) has been reported for decades and was not considered as paradoxical until recently.

L29 : you could mention here the study (46)

L31 : to be objective and complete you should mention also that some features of CH₄ in lakes cannot be explained by OMP. Such as under-ice CH₄ accumulation, or hypolimnion CH₄ accumulation.

L49 : you could mention here the study (46)

L 58 : you could add here tropical latitudes (46)

L84 and L198 : “excellent agreement” is a self-evaluation. Provide the stats of the comparison and let the reader decide the quality of the agreement.

Figure 2 : it could be useful to add in this figure or as a separate supplemental figure the bathymetry profile along these transects. Given these are alpine lakes, the bathymetry should be very steep, so probably the littoral zone extremely restricted. This will not be case of other type of lakes.

L 128 : Flick’s law describes molecular diffusion and not air-water fluxes. K600 does not have the dimension of a diffusion coefficient (m²/s) but of a velocity (m/s)

L 132: it’s usually called gas transfer velocity and not mass transfer velocity.

L 261: The eutrophic lakes BER and NOI also have much higher hypolimnetic CH₄ concentrations. So the eutrophy promotes sediment CH₄

L 267: you could mention here the study (46)

L283: This was demonstrated before by (46) who reported a relation between OMP and primary production. But in this case OMP was directly and accurately measured with stable isotope tracers rather than computed indirectly by mass balance with a large uncertainty.

Figure 6: Peeters and Hoffman (2021) recently showed that the estimates of Guenther et al. (2019) and Donis et al. (2017) were in fact wrong due to errors in terms of the mass balance. Are the values in this graph the original (and erroneously over-estimated) values or the corrected values proposed by Peeters and Hoffman (2021)?

The linear relation shown in Fig. S13 is not be statistically valid (specially given that the data have been log transformed). With 8 data points a linear relation is valid at 0.05 level for r² higher than 0.5 which is not the case here (r²=0.37). This figure is statistically meaningless and is not acceptable for a journal such as Nature Comm. Figure should be removed, as well as the accompanying text.

L 285: Statement “Methylphosphonate (MPn) biodegradation has been shown to be responsible for CH₄ production in oxic waters of the ocean (...)” is incorrect. What the Karl et al. (2008) showed is that if you add MPn to oligotrophic seawater then CH₄ appears during the incubation. While this is an interesting finding it’s hardly bullet-proof evidence of a process “responsible for CH₄ production in oxic waters of the ocean”. The main objection being that MPn does not occur naturally, it’s an artificial molecule for industrial applications.

L302: you could mention here the study (46)

L304: light inhibition of MOX was directly quantified by (46)

The relation in Figure 6 are mainly driven by the data in the four alpine lakes reported here, the data from the other lakes cluster close to the origin. So the relation mainly explains the data in 4 alpine lakes. A relation that explains variations in 4 lakes hardly makes any progress "towards a global scaling" as claimed in the title.

REFERENCES

Belanger, T. V., and E. A. Korzum. 1991. Critique of floating dome technique for estimating reaeration rates. *J. Environ. Eng.* 117: 144–150.

Peeters F & H. Hofmann (2021) Oxidic methanogenesis is only a minor source of lake-wide diffusive CH₄ emissions from lakes. *Nature Communications*, 12:1206, <https://doi.org/10.1038/s41467-021-21215-2>

Vachon D, YT Prairie and JJ Cole (2010) The relationship between near-surface turbulence and gas transfer velocity in freshwater systems and its implications for floating chamber measurements of gas exchange, *Limnology and Oceanography*, 55, 1723-1732

Reviewer #3 (Remarks to the Author):

Please find my comments in the attached file.

Reviewer #4 (Remarks to the Author):

Ordóñez et al. present a very interesting dataset suggesting that oxidic methane production occurs in the studied lakes. The authors conclude this from the results of both a full-scale mass balance and a lateral transport model to analyze sources and sinks of methane in the surface mixed layer of four pre-alpine lakes in the Swiss Alps. The study is well written, sections are generally clearly structured and the graphical presentation is outstanding. Discussion and conclusions are refreshingly kept cautious but straight forward and focus on the strengths and shortcomings of the used models and its parameters.

By this, I highly recommend this study for publication after revisions based on the comments, I listed below.

General comments

- The study lacks any information about the calculation (or determination?) of methane oxidation. These data are, however, mandatory for the calculation of PNet.
- Surface CH₄ concentrations were sampled along transects in each lake and by this, CH₄ concentration was simulated using the lateral transport model. However, Figure 2 shows large deviations for near-shore concentrations. These deviations should be mentioned clearly. Although PNet is calculated in the center of the lake, the authors should state if & why the flux from the sediment is generally underestimated by their full-lake model and how this affects the overall mass balance.

Specific comments

- Line 111: Add SML in "along a transect from shore to shore"
- Line 113: When was the difference in oversaturation significant between shore and center of the lake?
- Fig 2: add information about outliers in the caption
- Line 128-148: Consider to move method specific information to material & methods section
- Line 163-165: Consider to move method specific information to material & methods section

- **Line 171-174: Consider to move method specific information to material & methods section**
- **Line 185-196: Consider to move method specific information to material & methods section**
- **Line 239: Add information on trophic state**
- **Fig 5: Add "eutrophic" and "oligotrophic" similar to Fig 3**
- **Consider calculating R^2 for oligotrophic and eutrophic individually. Is there any significant difference?**
- **Line 309: Refer to Fig 6**
- **Fig 7: Add "eutrophic" and "oligotrophic" similar to Fig 3**
- **Line 475: add information on injected volume**
- **Line 476: correct to Picarro G2201-i**

Reviewer's Comments:

Reviewer #1 (Remarks to the Author):

MAJOR COMMENTS

The authors present in an attractive way CH₄ data reported in 4 alpine lakes. They compute to close a
mass balance what they call net CH₄ production (P_{net}) corresponding to oxic methane production
(OMP) minus methane oxidation (MOX). They then use their data with another two studies to derive a
general relation between P_{net} and chlorophyll-a concentration (plus light regime indicated by Secchi
depth). This relation is supposed to contribute "towards a global scaling of oxic methane production" as
claimed in the title.

There are several problems in the approach.

First, P_{net} is computed as an input term to close the mass balance of several input/output terms. This
means that if an output term is over-estimated, mathematically P_{net} is automatically over-estimated.
This seems to be the case for the diffusive emission of CH₄ to the atmosphere derived from floating
chambers that are known to provide over-estimates of gas emissions (This is quite intuitive if you
picture mentally a chamber being thrown around by waves or dragged by wind).

Alternatively if another input term is under-estimated, mathematically P_{net} is also automatically over-
estimated. This seems to be the case of CH₄ input in the mixed layer from dissolution of CH₄ from
rising bubbles (originating from sedimentary ebullition). CH₄ ebullition was not actually measured but
seems to have been computed from sediment data in a way that is briefly mentioned in text in an
obscure way. So it is possible (and in my opinion very likely) that the reported P_{net} are vastly over-
estimated.

Second, the authors attempt to derive a "general" relation between P_{net} and Chlorophyll and light
regime. The problem is that P_{net} is normalized by CH₄. On the one end, there is no conceptual reason
for such a normalization, in fact it does not make sense at all to do such a normalization. On the other
hand it's the normalization itself that most probably drives the relationship, and that non-normalized
P_{net} is unrelated to Chlorophyll and light regime.

Third, to derive a "general" relation between P_{net} and Chlorophyll, the authors use the data of
Guenther et al. (2019) and Donis et al. (2017) that were shown recently by Peeters and Hoffman (2021)

to actually be erroneously over-estimated. I find it puzzling that authors used discredited data in their
analysis.

Fourth, to derive a “general” relation between Pnet and Chlorophyll, the authors have excluded the
independent study of Morana et al. (ref 46) that reported data that could have been used in attempting
to build a general relation between Pnet and Chlorophyll and light regime.

137-141 : I find it very worrying that chamber estimates of K600 are over-estimated compared to 5
independent parameterizations. For me this is indicative that the measurements reported by the authors
are over-estimated due to a methodological bias. And indeed, chamber measurements are well known
to over-estimate fluxes due to pressure and temperature changes in the chamber during the deployment
(Belanger and Korzum 1991) or to artificial enhancement of water turbulence due to the movement of
the chamber (Vachon et al. 2010). The movement of the chamber can be due to the oscillation of the
chamber with waves or the chamber being dragged by the wind. What is also worrying is that the
authors do not present any convincing physical explanation for the fact that these 4 lakes would be so
special as to have higher k600 values. For instance the Cole and Caraco parameterization is based on
the compilation of numerous tracer experiments in numerous lakes of variable surface area and depth.
So this parameterization should cover the typical range of physical attributes of lakes

This issue is not marginal, since net OMP is calculated (as an input) by difference in a mass balance in
which the diffusive emission to the atmosphere is an output term. If one the output term is erroneously
over-estimated, automatically net OMP is also erroneously over-estimated since it is computed to close
the mass balance.

In a similar note, an important input of CH₄ to the oxygenated mixed layer is the dissolution of CH₄
from the rising bubbles from the sediment. This is an important process because it allows CH₄
produced in the sediments to enter the oxygenated mixed layer, bypassing water column methane
oxygenation. This process alone might in theory explain all of the reported OMP that is calculated to
close a mass balance.

For instance the reported higher OMP in the two eutrophic lakes (BER and NOI) would be consistent
with a higher ebullition in these lakes (higher sediment organic matter) and dissolution of CH₄ in the
oxygenated mixed layer from rising bubbles. The problem is that it is unclear from the ms how the
ebullitive fluxes were actually derived. From text in line 156, it seems that ebullitive fluxes were
somehow modelled from diffusive fluxes. It would useful if the authors could explain in much more

detail how exactly this was achieved, and what would be error associated to this ebullitive fluxes.
Similarly it would be useful if they provide an error propagation estimate on OMP (as done for the air-
water estimates).

Available studies show that OMP is linked to phytoplankton metabolism (papers from Grossart, Bizic,
etc). The exact nature of the this link is not fully resolved and as rightly pointed out the authors it could
be multiple and variable from one lake to another. Yet, whatever the mechanism there should be link
between phytoplankton biomass and OMP. This was shown in the (selection of) vertical profiles shown
in the Grossart et al. (2011) paper, showing a relation between the peaks of CH₄, Chlorophyll-a and O₂
at the thermocline in lake Stechlin. I'm surprised that the chlorophyll-a profiles from the lakes are not
shown in Figure S2. On the contrary to what would be expected, the lack of relation between O₂ and
CH₄ suggest there are no functional relation as would be expected from OMP and primary production.
On the contrary, the vertical structure of the concentration of CH₄ and of the stable isotope of CH₄
show "classical" patterns that can be interpreted as resulting from the inputs of CH₄ from sediments
(diffusion or dissolution of bubbles).

306: I would like to see the graph of CH₄ concentration in the SML versus Chla X LC x Secchi depth.
If CH₄ correlates given that CH₄ typically changes over 1 or 2 orders of magnitude in the SML, the
relation between Pnet and Chla X LC x Secchi depth could simply result from the normalization.

Also, from a conceptual point of view this normalization is very strange. Trophic status correlates with
Chla (and light). Since Pnet correlates to trophic status (as stated), then non-normalized Pnet should
correlate to Chla X LC x Secchi. So why did you normalize Pnet for trophic status ? This simply does
not make sense from a conceptual point of view. The only use of such an approach would be if CH₄
strongly relates to Chla X LC x Secchi, and the normalization allows to artificially derive a relationship
between normalized Pnet and Chla X LC x Secchi.

L 309: Statement "While more data are needed to understand this parameterizaion" is intriguing.
Morana et al. (2021) (ref 46) reported OMP and MOX (allowing to compute Pnet) as well as Chla and
Secchi depth. This was reported in 3 lakes. This would allow to substantially increase the data-set given
the actually low number of data points used by the authors to derive their relation. Also the work of
Morana spans a large gradient of productivity and in a tropical climate that would allow to really
contribute "towards a global scaling".

MINOR COMMENTS

L25-26 : « methane paradox » concept has only been recently extended to lakes. Over-saturation of
CH₄ in surface waters of lakes (that are oxic) has been reported for decades and was not considered as
paradoxical until recently.

L29 : you could mention here the study (46)

L31 : to be objective and complete you should mention also that some features of CH₄ in lakes cannot
be explained by OMP. Such as under-ice CH₄ accumulation, or hypolimnion CH₄ accumulation.

L49 : you could mention here the study (46)

L 58 : you could add here tropical latitudes (46)

L84 and L198 : “excellent agreement” is a self-evaluation. Provide the stats of the comparison and let
the reader decide the quality of the agreement.

Figure 2 : it could be useful to add in this figure or as a separate supplemental figure the bathymetry
profile along these transects. Given these are alpine lakes, the bathymetry should be very steep, so
probably the littoral zone extremely restricted. This will not be case of other type of lakes.

L 128 : Flick’s law describes molecular diffusion and not air-water fluxes. K600 does not have the
dimension of a diffusion coefficient (m²/s) but of a velocity (m/s)

L 132: it’s usually called gas transfer velocity and not mass transfer velocity.

L 261: The eutrophic lakes BER and NOI also have much higher hypolimnetic CH₄ concentrations. So
the eutrophy promotes sediment CH₄

L 267: you could mention here the study (46)

L283: This was demonstrated before by (46) who reported a relation between OMP and primary
production. But in this case OMP was directly and accurately measured with stable isotope tracers
rather than computed indirectly by mass balance with a large uncertainty.

Figure 6: Peeters and Hoffman (2021) recently showed that the estimates of Guentzel et al. (2019) and
Donis et al. (2017) were in fact wrong due to errors in terms of the mass balance. Are the values in this
graph the original (and erroneously over-estimated) values or the corrected values proposed by Peeters
and Hoffman (2021)?

The linear relation shown in Fig. S13 is not be statistically valid (specially given that the data have
been log transformed). With 8 data points a linear relation is valid at 0.05 level for r² higher than 0.5

which is not the case here ($r^2=0.37$). This figure is statistically meaningless and is not acceptable for a
journal such as Nature Comm. Figure should be removed, as well as the accompanying text.

L 285: Statement “Methylphosphonate (MPn) biodegradation has been shown to be responsible for
CH₄ production in oxic waters of the ocean (...)” is incorrect. What the Karl et al. (2008) showed is
that if you add MPn to oligotrophic seawater then CH₄ appears during the incubation. While this is an
interesting finding it’s hardly bullet-proof evidence of a process “responsible for CH₄ production in
oxic waters of the ocean”. The main objection being that MPn does not occur naturally, it’s an artificial
molecule for industrial applications.

L302: you could mention here the study (46)

L304: light inhibition of MOX was directly quantified by (46)

The relation in Figure 6 are mainly driven by the data in the four alpine lakes reported here, the data
from the other lakes cluster close to the origin. So the relation mainly explains the data in 4 alpine
lakes. A relation that explains variations in 4 lakes hardly makes any progress “towards a global
scaling” as claimed in the title.

REFERENCES

Belanger, T. V., and E. A. Korzum. 1991. Critique of floating dome technique for estimating reaeration
rates. J. Environ. Eng. 117: 144–150.

Peeters F & H. Hofmann (2021) Oxidic methanogenesis is only a minor source of lake-wide diffusive
CH₄ emissions from lakes. Nature Communications, 12:1206, [https://doi.org/10.1038/s41467-021-](https://doi.org/10.1038/s41467-021-21215-2)
[21215-2](https://doi.org/10.1038/s41467-021-21215-2)

Vachon D, YT Prairie and JJ Cole (2010) The relationship between near-surface turbulence and gas
transfer velocity in freshwater systems and its implications for floating chamber measurements of gas
exchange, Limnology and Oceanography, 55, 1723-1732

**Reviewer #2 (Remarks to the Author):**

Ordóñez et al. present a very interesting dataset suggesting that oxidic methane production occurs in the
studied lakes. The authors conclude this from the results of both a full-scale mass balance and a lateral

transport model to analyze sources and sinks of methane in the surface mixed layer of four pre-alpine
lakes in the Swiss Alps.

The study is well written, sections are generally clearly structured and the graphical presentation is
outstanding. Discussion and conclusions are refreshingly kept cautious but straight forward and focus
on the strengths and shortcomings of the used models and its parameters.

By this, I highly recommend this study for publication after revisions based on the comments, I listed
below.

General comments:

The study lacks any information about the calculation (or determination?) of methane oxidation. These
data are, however, mandatory for the calculation of PNet.

Surface CH₄ concentrations were sampled along transects in each lake and by this, CH₄ concentration
was simulated using the lateral transport model. However, Figure 2 shows large deviations for near-
shore concentrations. These deviations should be mentioned clearly. Although PNet is calculated in the
center of the lake, the authors should state if & why the flux from the sediment is generally
underestimated by their full-lake model and how this affects the overall mass balance.

Specific comments

- • Line 111: Add SML in “along a transect from shore to shore”
- • Line 113: When was the difference in oversaturation significant between shore and center of the
lake?
- • Fig 2: add information about outliers in the caption
- • Line 128-148: Consider to move method specific information to material & methods section
Moved
- • Line 163-165: Consider to move method specific information to material & methods section
- • Line 171-174: Consider to move method specific information to material & methods section.
- • Line 185-196: Consider to move method specific information to material & methods section.
Moved to methods
- • Line 239: Add information on trophic state.

- • Fig 5: Add “eutrophic” and “oligotrophic” similar to Fig 3.
- • Consider calculating R^2 for oligotrophic and eutrophic individually. Is there any significant
difference?
- • Line 309: Refer to Fig 6.
- • Fig 7: Add “eutrophic” and “oligotrophic” similar to Fig 3.
- • Line 475: add information on injected volume
- • Line 476: correct to Picarro G2201-i. We had corrected to G2201-i

**Reviewer #3 (Remarks to the Author):**

General comments

The subject of the manuscript is relevant since oxic methane production is a little-known process, and
its investigation may help reduce uncertainties in large-scale methane flux budgets. The introduction is
well written and reflects the importance of the subject. The methods used are suitable, and authors have
clearly put in a lot of effort in producing a valuable and extensive dataset. Overall, this is a nice and
important study. However, I believe some major aspects of the paper need to be improved prior to
publication, mainly concerning the structure, interpretation, and focus of the discussion:

- • In terms of structure, the “Results” and “Methods” section are very confusing since a large
portion of the methodological information (including major equations) is found in the “Results”
section rather than the methods, making the manuscript hard to follow. Methodological and
results information are also found in the discussion section (ex. Line 321-334). Authors should
work on properly separating sections and avoid mixing the different types of information.
- • The discussion section is largely focused on justifying the validity of the P_{net} estimates as and
contrasting it to Peeters et al. study. While justifying methods and backing up observed results
is important, it should not be the main focus of the discussion. Despite the original paradigm
and debate associated with oxic methane production, many recent studies have shown, using
different approaches, the existence and importance of this process, and it is now an established
methane source. Authors could instead expand the discussion on potential OMP drivers, which

is rather limited in the manuscript. For instance, further explore results, analyses, and
hypotheses on seasonal and inter-lake patterns of OMP rate and contribution to flux.

- • In the discussion, authors mention that uncertainty on Pnet estimates are largely related to
uncertainty in K600. This is in itself a novel and important result for this study and future
assessments of OMP. However, results of the uncertainty calculation (sensitivity analysis) are
not clearly shown. It would be useful to mention in the results / figures / discussion what is the
actual change in OMP caused by a certain error on k600. This is especially important given that
both measured and modeled k600 values have associated error. Indeed, measured k600 is
restricted to a 5-min time scale, not necessarily representative of diel patterns. This limitation of
measured k600 should also be discussed in the choice of method and interpretation of the data.
In terms of OMP uncertainty it would also be interesting to present values on the sensitivity of
OMP estimates to factors other than k600 as well for comparison. This would yield novel and
constructive information in terms of advancing OMP estimation in a larger context.
- • The manuscript contains 7 figures and 3 tables in the main text, with an additional 14 figures
and 5 tables in the supplementary information. This number of displays is unreasonable, and
reflects a general lack of structure in the study, failing to summarize and focus on important
information and target specific research questions (especially in the discussion). Authors should
drastically reduce the number of displays based on the main messages (conclusions) they want
to convey.

Specific comments

- • Isotopic signature data are presented in the results but never mentioned in the discussion. Thus,
they should be either removed from the manuscript or interpreted in the discussion if they
provide valuable insights.
- • Line 12: more widely accepted than what? Be specific.
- • Line 20: at this point it is not „newly recognized“ given historic marine research. Revise.
- • Line 22: This statement is alarmist and must be changed, A positive feedback loop assumes that
oxic lake CH₄ emissions will deliver enough CH₄ due to eutrophication/climate change (above
and beyond baseline oxic fluxes) that it significantly drives the atmospheric CH₄ concentration

up. The scale of the flux we are talking about here is nowhere near that large, and the
mechanistic evidence for the statement is not available. Maybe refocus conclusion sentence to
emphasize the importance of the dataset and modelling technique and how it advances
understanding of the lake CH₄ cycle.

- • Line 33: since onset of industrial era?
- • Line 43: it is quickly
- • Line 58: altitude
- • Line 71: “with identical climate” change “with” to “facing”
- • In the description of the model (ex. Line 72 - 74), the formulation sound like OMP is the only
methane source considered in the model. Please clarify that other sources are accounted and
OMP is not the only process “that adds CH₄” (line 73).
- • Line 85: add the actual numbers for the contribution of OMP
- • Line 113: Don’t agree with this artificial breakdown without some proof up front that each lake
falls in the trophic status categories they are assigned to. This applies here and throughout.
- • Line 121: color in this graph and others is not distinct enough in greyscale.
- • Line 131-132: add reference for the exponent
- • Line 148: Hypereutrophic? How is this possible?
- • Line 152: less enriched, not “lighter” – use proper terminology.
- • Line 200-203: the values in the two sentences do not seem to match.
- • Line 230: But I count 6 points below the 1:1 line in fig. 4c. Is this statement correct?
- • Line 245: Is the comparison in fig 4b showing a circular relationship? Is the chamber-based flux
contained in both x and y axis metrics? Confirm this is not a problem with a quick sensitivity
analysis?
- • Line 251-253: is there a link between Pnet and water level?
- • Line 256: this is a weird sentence formulation. Within or across sites?
- • Line 257 to 265: Since Pnet and MO_x not directly confirmed here, please be more cautious in
these statements as the mass-balance essentially remains unvalidated.

- • Line 267: Disagree, it is not hotly debated. One group disagrees, but the vast majority of papers
show some kind of support and converge to an agreement that OMP cycling is relevant to
surface water processes.
- • Line 274 to 276: Seems like an obvious statement that could be replaced with a more
informative conclusion.
- • Line 286 to 288: Wouldn't P limitation depend more on N to P ratios, not only [P]?
- • Line 294: Also this relationship is shown in reference 6. Could cite as further support.
- • Line 299-301: I don't quite understand this approach, please clarify.
- • Line 305-307: I don't understand the choice of these parameters. Why include both Chla and
light parameters if light only influences Pnet indirectly through Chla? Also, this statistical
analysis is not described in the methods or results, it is a multiple regression? What is the
equation? Is it presented in a figure?
- • Line 309-312: I disagree with the authors that this particular result provides "a step towards
estimating Pnet" as they provide no numerical equation, and the link between OMP and Chla is
already known.
- • Line 333 to 339: Rethink this interpretation? Day-time chambers are selecting for the most
turbulent time of day, so it makes sense that other models will have a lower (possibly more
representative) k_{600} ?
- • Line 350: Does this take into consideration the fact that day-time measurements don't account
for night-time physical conditions that are often quite different?
- • Line 381: "need it" should be "needed"?
- • Line 421: the term mass balance is confusing here because it refers to both the full-scale mass
balance and the lateral transport model if I understand correctly?
- • Line 438: "horizontal" should be "horizontal plane"?
- • Line 478: specify what is the headspace/water ratio used.
- • Table S2: showing means and SD or medians may be more informative than just the range here.
- • Table S2: It is weird that the dates are not shown in order (Sep. before July).

Point-by-point response to the reviewer's comments on the
manuscript: "Towards a global scaling of oxic methane production:
evaluation of the methane paradox in four Swiss pre-alpine lakes" by
Ordóñez and colleagues

**1 Reviewer #1 (Remarks to the Author):**

**1 Major comments**

The authors present in an attractive way CH₄ data reported in 4 alpine lakes. They compute to close a
mass balance what they call net CH₄ production (P_{net}) corresponding to oxic methane production
(OMP) minus methane oxidation (MOX). They then use their data with another two studies to derive a
general relation between P_{net} and chlorophyll-a concentration (plus light regime indicated by Secchi
depth). This relation is supposed to contribute "towards a global scaling of oxic methane production" as
claimed in the title.

There are several problems in the approach.

First, P_{net} is computed an input term to close the mass balance of several input/output terms. This
means that if an output term is over-estimated, mathematically P_{net} is automatically over-estimated.
This seems to be the case for the diffusive emission of CH₄ to the atmosphere derived from floating
chambers that are known to provide over-estimates of gas emissions (This is quite intuitive if you
picture mentally a chamber being thrown around by waves or dragged by wind).

**Response:** We respect the reviewer's skepticism towards using floating chambers for air-water gas
exchange measurements, but we respectfully disagree with the reviewer on their applicability.
Chambers have been found to be a reliable tool for measuring gas fluxes (see e.g., Cole et al. 2010;
Perolo et al. 2021; Matthews et al. 2003). A key issue is that for small lakes (<50 ha) k_{600} is reportedly
less dependent on wind for low wind speeds (between 2 – 3 m/s) and thus it becomes more important to
obtain a lake-specific k_{600} range (Cole et al. 2010 and references therein).

There are many studies addressing the applicability of floating chambers to resolve fluxes. For
example, Cole et al. (2010) specifically compare the chamber with tracer methods for resolving k_{600} .
The reviewer believes they are prone to overestimating flux, but this is only the case in flowing water
when chambers themselves are causing extra turbulence (Lorke et al. 2015). If chambers are not
causing excess turbulence, then they are in fact more likely to underestimate flux as they block the
wind from impacting the air-water interface within the chamber and reduce evaporative cooling (by
trapping water vapor) that causes small-scale convection which enhances the mass transfer coefficient k
(see MacIntyre et al., 2010). Lorke et al. (2015) looked at the effect of dragging the chamber (or a
stationary chamber mounted in a flowing stream), and as expected this does enhance the k values due
to enhanced turbulence at the edges of the chamber, but concluded experimentally that if the chamber is
stationary relative to the water then this is negligible. In our study, we deployed these chambers on
small, relatively wind-protected lakes where there were no significant waves (i.e., smooth water
surface) or dragging by wind (same flux chamber as in McGinnis et al. 2015). We are confident in this
methodology as a robust means to resolve CH_4 and CO_2 fluxes.

Alternatively if another input term is under-estimated, mathematically P_{net} is also automatically over-
estimated. This seems to be the case of CH_4 input in the mixed layer from dissolution of CH_4 from
rising bubbles (originating from sedimentary ebullition). CH_4 ebullition was not actually measured but
seems to have been computed from sediment data in a way that is briefly mentioned in text in an
obscure way. So it is possible (and in my opinion very likely) that the reported P_{net} are vastly over-
estimated.

**Response:** If ebullition was underestimated then P_{net} could be overestimated. However, we accounted
for this process in a relatively conservative manner. Firstly, we used an echosounder in each lake to
survey for ebullition following DelSontro et al. (2015). We found no ebullition in two of the lakes
(CHA and LIO) and only minimal bubbling in BRE and NOI. Though we could hydroacoustically
observe bubbles, we were unable to estimate ebullition with this method in those two lakes due to the
shallowness and the macrophytes in the littoral zones where the bubbles were mostly observed.
Therefore, we used a published sediment model from Langenegger et al. (2019) to estimate ebullition
in these lakes. The model is based on sampling sediment bubble gas content. Together with the
measured fluxes at the sediment/water interface, the ebullition is estimated using inverse modeling. We
have included more details regarding this method in the text (Lines 568-576). Note that the ebullitive

fluxes are within the range of ebullition in the literature for these types of lakes ($\lesssim 6 \text{ mmolm}^{-2}\text{d}^{-1}$)
(Rinta et al., 2017, Vachon et al., 2017, DelSontro et al., 2016).

In addition, we conducted many flux chamber measurements in transects along the lakes during each
campaign and rarely did we encounter bubbles in our chambers. Note that the overall contribution from
bubbling to the surface mixed layer is generally very low because the contact time between the bubble
and ambient water is extremely short (~10 seconds), and thus dissolution is very low. We added a
sensitivity analysis in discussion (lines 318-330) and SI (lines 883-887), where we showed that if we
do not consider OMP in BRE and NOI, we would need one or two order of magnitude higher ebullitive
fluxes than what we estimated to close the mass balance and what it is usually found in these type of
lakes (Rinta et al. 2017, Vachon et al 2019, DelSontro et al. 2016).

Second, the authors attempt to derive a “general” relation between Pnet and Chlorophyll and light
regime. The problem is that Pnet is normalized by CH₄. On the one end, there is no conceptual reason
for such a normalization, in fact it does not make sense at all to do such a normalization. On the other
hand it’s the normalization itself that most probably drives the relationship, and that non-normalized
Pnet is unrelated to Chlorophyll and light regime.

**Response:** We thank the reviewer for this comment. In this response, we would like to address two
related comments raised by the reviewer written above:

- 1. 306: I would like to see the graph of CH₄ concentration in the SML versus Chla X LC x Secchi
depth. If CH₄ correlates given that CH₄ typically changes over 1 or 2 orders of magnitude in
the SML, the relation between Pnet and Chla X LC x Secchi depth could simply result from the
normalization. Also, from a conceptual point of view this normalization is very strange. Trophic
status correlates with Chla (and light). Since Pnet correlates to trophic status (as stated), then
non-normalized Pnet should correlate to Chla X LC x Secchi. So why did you normalize Pnet
for trophic status ? This simply does not make sense from a conceptual point of view. The only
use of such an approach would be if CH₄ strongly relates to Chla X LC x Secchi, and the
normalization allows to artificially derive a relationship between normalized Pnet and Chla X
LC x Secchi.”
- 2. L 309: Statement “While more data are needed to understand this parameterizaion” is intriguing.
Morana et al. (2021) (ref 46) reported OMP and MOX (allowing to compute Pnet) as well as

Chla and Secchi depth. This was reported in 3 lakes. This would allow to substantially increase
the data-set given the actually low number of data points used by the authors to derive their
relation. Also the work of Morana spans a large gradient of productivity and in a tropical
climate that would allow to really contribute “towards a global scaling”.

We have put a lot of thought behind this and have included a new figure panel to Fig 6 (new Fig 5)
where we observe a relationship between P_{net} and light climate (LC). In Fig 5a, we observed that in the
eutrophic lakes P_{net} strongly depends on LC while for the oligotrophic lakes it becomes almost
independent. Since eutrophic lakes are generally more turbid (due to algal biomass) than oligotrophic
lakes, we would expect to observe greater changes of the average light intensities in the SML. We
hypothesized that the relation between P_{net} and LC are related with ROS production at high light
intensities that could potentially increase OMP (Ernst et al., 2022) and also MOx inhibition.

This result shows that despite the fact that different phytoplankton species are able to produce CH_4
under oxic conditions, environmental conditions such as LC are key to control the CH_4 production rates
by phytoplankton. This could explain the lack of correlation between P_{net} and Chla in our lakes.
However, since this relation is trophic state dependent, we included trophic state variables (CH_4 surface
concentrations and SD) to build our upscaling method (Fig 5b).

Figure 5. **a** Interaction between P_{net} ($\text{mmol m}^{-3} \text{d}^{-1}$) and light climate (LC, m m^{-1}) and trophic state. Per lake, the
 minimum Pnet rate ($P_{net, min}$) and the minimum LC were subtracted to be able to compare the slope of each
 curve. On panel **a** we had included only the lakes with more than one data points (BRE, NOI, CHA, LIO and
 Stechlin). **b** Interaction between P_{net} ($\text{mmol m}^{-3} \text{d}^{-1}$) and $Chla$ (mg m^{-3}), LC (-) and SD (m) suggest a direct role of
 photosynthesis on OMP. Specific production/oxidation rate calculated as P_{net} normalized by CH_4 concentration
 (mmol m^{-3}) ($P_{net} C_{CH_4}^{-1}$) versus $Chla \times$ light climate ($LC = 2.5SD/H_{SML}$) SD. C_{CH_4} is the average surface
 concentrations, $Chla$ is the surface average concentrations obtained from CTD's profiles at the center of the lake,
 SD is the Secchi depth and H_{SML} is the SML depth. All the parameters were calculated at each sampling
 campaign.

CH_4 concentrations (and often CH_4 emissions) are dependent on trophic state where higher CH_4
 concentrations are generally observed in eutrophic lakes and lower in oligotrophic lakes (DelSontro et
 al., 2016). Therefore, we suggest that CH_4 concentration can be utilized as a trophic indicator (in
 addition to $Chla$ and Secchi depth). Thus we use the standardization of P_{net} with CH_4 to scale across
 lakes of different trophic states. We clarified this in the text lines 354-379. As requested by the
 reviewer, in the figure below (only here) we showed that there is no correlation between CH_4
 concentration versus chlorophyll-a, light climate and Secchi depth ($R^2 = 0.01$) as a proof that the

normalization by CH₄ concentration is not artificially creating a relationship between P_{net} and Chla x
LC x SD.

Figure: Correlation between surface CH₄ concentration versus Chla x LC x SD.

Finally, we have added five new lakes at our upscaling method (new Fig. 5b) from Thottathil et al.,
(2022) where P_{net} rates were calculated using a mass balance approach. We observed almost no change
in our result, the correlation coefficient decreased from R² = 0.85 to R² = 0.79. Please see response
below (lines 419-421 on this document) where we explain why we did not include Morana et al.,
(2021) in our scaling Figure 6 (new Fig. 5b).

Third, to derive a “general” relation between P_{net} and Chlorophyll, the authors use the data of
Guenthel et al. (2019) and Donis et al. (2017) that were shown recently by Peeters and Hoffman (2021)
to actually be erroneously over-estimated. I find it puzzling that authors used discredited data in their
analysis.

**Response:** The reviewer is not correct as we did not use discredited data in our analysis. Peeters et al.
(2019) correctly point out that Donis et al. (2017) used incorrect bathymetry for their estimates;
however, the P_{net} estimates used in Günthel et al. (2019) were based on the corrected bathymetry. The
new resulting P_{net} was slightly less than the original estimate in Donis et al. (2017) but its contribution
to diffusive emissions was about 63% (Günthel et al., 2019). Perhaps the reviewer refers to the
response from Peters & Hofmann (2021) that claimed that Donis et al. (2017) and Günthel et al., 2019
miscalculated K's. However, we point the reviewer to the rebuttal to Peters & Hofmann (2021) by

Günthel et al. (2021) that the fluxes used in the mass balance are indeed correct. We strongly urge the
reviewer to carefully read the response of Günthel et al. (2021) before incorrectly stating that the data
from Günthel and Donis are “discredited”. <https://doi.org/10.1038/s41467-021-21216-1>

Fourth, to derive a “general” relation between Pnet and Chlorophyll, the authors have excluded the
independent study of Morana et al. (ref 46) that reported data that could have been used in attempting
to build a general relation between Pnet and Chlorophyll and light regime.

**Response:** Even though Morana et al. (2021) estimate OMP independently, unfortunately we had to
exclude Morana et al. (2021) because the mass balance in their lakes does not close (for example in
Lake George, a source equivalent to about 50% of its losses is missing).

137-141 : I find it very worrying that chamber estimates of K600 are over-estimated compared to 5
independent parameterizations. For me this is indicative that the measurements reported by the authors
are over-estimated due to a methodological bias. And indeed, chamber measurements are well known
to over-estimate fluxes due to pressure and temperature changes in the chamber during the deployment
(Belanger and Korzum 1991) or to artificial enhancement of water turbulence due to the movement of
the chamber (Vachon et al. 2010). The movement of the chamber can be due to the oscillation of the
chamber with waves or the chamber being dragged by the wind. What is also worrying is that the
authors do not present any convincing physical explanation for the fact that these 4 lakes would be so
special as to have higher k600 values. For instance the Cole and Caraco parameterization is based on
the compilation of numerous tracer experiments in numerous lakes of variable surface area and depth.
So this parameterization should cover the typical range of physical attributes of lakes.

This issue is not marginal, since net OMP is calculated (as an input) by difference in a mass balance in
which the diffusive emission to the atmosphere is an output term. If one the output term is erroneously
over-estimated, automatically net OMP is also erroneously over-estimated since it is computed to close
the mass balance.

**Response:** If we overestimate the flux to the atmosphere we would indeed overestimate OMP and
similarly, if we overestimate flux from the sediment we would underestimate OMP. However, we argue

that chambers are a reliable tool for measuring gas fluxes (please see our previous response to the first
major point on lines 296-315 in this document). To keep the budget as accurate as possible, the k_{600s}
need to be resolved on a per-lake basis as suggested in Cole et al. (2010).

In a similar note, an important input of CH₄ to the oxygenated mixed layer is the dissolution of CH₄
from the rising bubbles from the sediment. This is an important process because it allows CH₄
produced in the sediments to enter the oxygenated mixed layer, bypassing water column methane
oxygenation. This process alone might in theory explain all of the reported OMP that is calculated to
close a mass balance.

**Response:** We realize this and its addressed in the manuscript including a sensitivity analysis (Lines
318-330 on the discussion section and 883-887 in the SI). Please see our previous response on lines
322-341 on this document.

For instance the reported higher OMP in the two eutrophic lakes (BER and NOI) would be consistent
with a higher ebullition in these lakes (higher sediment organic matter) and dissolution of CH₄ in the
oxygenated mixed layer from rising bubbles. The problem is that it is unclear from the ms how the
ebullitive fluxes were actually derived. From text in line 156, it seems that ebullitive fluxes were
somehow modelled from diffusive fluxes. It would be useful if the authors could explain in much more
detail how exactly this was achieved, and what would be the error associated to this ebullitive fluxes.
Similarly it would be useful if they provide an error propagation estimate on OMP (as done for the air-
water estimates).

**Response:** We have expanded the description (Lines 557-576), and we refer readers to the open access
paper by Langenegger et al. (2019), where this approach is very thoroughly described. As described in
lines 186, 470-477, to calculate the P_{net} rates, we conducted Monte Carlo simulations (N=10000) to
include the variability of all the inputs in our mass balance models (bubble dissolution, littoral sediment
flux, diffusive emissions and vertical transport) (see Methods lines 470-477).

Available studies show that OMP is linked to phytoplankton metabolism (papers from Grossart, Bizic,
etc). The exact nature of this link is not fully resolved and as rightly pointed out by the authors it could

be multiple and variable from one lake to another. Yet, whatever the mechanism there should be link
between phytoplankton biomass and OMP. This was shown in the (selection of) vertical profiles shown
in the Grossart et al. (2011) paper, showing a relation between the peaks of CH₄, Chlorophyll-a and O₂
at the thermocline in lake Stechlin. I'm surprised that the chlorophyll-a profiles from the lakes are not
shown in Figure S2. On the contrary to what would be expected, the lack of relation between O₂ and
CH₄ suggest there are no functional relation as would be expected from OMP and primary production.

**Response:** We frankly find this comment puzzling. The “peaks” observed in Grossart et al. (2011) and
also shown in Donis et al. (2017) that overlap with the Chla, while they may appear dramatic, have a
very low OMP rate associated with them. This is because they occur in the thermocline where the
turbulent vertical diffusivity and light are very low compared to the surface mixed layer. It is crucial to
evaluate rates in these cases as the concentration alone tells you very little about the functioning of the
system. We encourage the reader to carefully read the analysis in Donis et al. (2017) where these zones
(P_{net} in the surface layer vs. the thermocline) are compared (Fig. 6 in Donis et al. 2017).

On the contrary, the vertical structure of the concentration of CH₄ and of the stable isotope of CH₄
show “classical” patterns that can be interpreted as resulting from the inputs of CH₄ from sediments
(diffusion or dissolution of bubbles).

**Response:** We are not sure what the review refers to as “classical patterns” and how this can be used to
classify the functioning and rates in a lake and, therefore we respectfully disagree with the reviewer on
this point. The vertical profiles of CH₄ and $\delta^{13}C$ of CH₄ can provide a hint as to what the source of CH₄
is at that location but it is not possible to provide any definitive conclusion. The instantaneous
concentration is a result of several processes occurring that leave that remaining CH₄ concentration. For
example, input from the sediments, OMP, mixing through the water column, and methane oxidation. To
ascertain what led to the observed concentrations, the rates of these processes must be used in a model
or mass balance. This is also true of the $\delta^{13}C$ of CH₄, although it is slightly more complicated to derive
processes from only one isotope if fractionation occurs on other elements in the molecule as well.
Ultimately, merely looking a profile cannot provide the sort of information that we provide with a
comprehensive mass balance and modeling study.

306: I would like to see the graph of CH₄ concentration in the SML versus Chla X LC x Secchi depth.
If CH₄ correlates given that CH₄ typically changes over 1 or 2 orders of magnitude in the SML, the
relation between Pnet and Chla X LC x Secchi depth could simply result from the normalization.
Also, from a conceptual point of view this normalization is very strange. Trophic status correlates with
Chla (and light). Since Pnet correlates to trophic status (as stated), then non-normalized Pnet should
correlate to Chla X LC x Secchi. So why did you normalize Pnet for trophic status ? This simply does
not make sense from a conceptual point of view. The only use of such an approach would be if CH₄
strongly relates to Chla X LC x Secchi, and the normalization allows to artificially derive a relationship
between normalized Pnet and Chla X LC x Secchi.

**Response:** See response on lines 366-401 on this document.

L 309: Statement “While more data are needed to understand this parameterizaion”is intriguing.
Morana et al. (2021) (ref 46) reported OMP and MOX (allowing to compute Pnet) as well as Chla and
Secchi depth. This was reported in 3 lakes. This would allow to substantially increase the data-set given
the actually low number of data points used by the authors to derive their relation. Also the work of
Morana spans a large gradient of productivity and in a tropical climate that would allow to really
contribute “towards a global scaling”.

**Response:** We have included five additional lakes to our upscaling methods (see more details on
response on lines 397-401 on this document). See the response about why we did not include Morana et
al., 2021 on lines 419-421 on this document

**Minor comments**

L25-26 : « methane paradox » concept has only been recently extended to lakes. Over-saturation of
CH₄ in surface waters of lakes (that are oxic) has been reported for decades and was not considered as
paradoxical until recently. The reviewer is correct.

L29 : you could mention here the study (46). We already have six references here.

L31 : to be objective and complete you should mention also that some features of CH₄ in lakes cannot
be explained by OMP. Such as under-ice CH₄ accumulation, or hypolimnion CH₄ accumulation.

We respectfully disagree with the reviewers, as we are focusing of OMP at the surface oxic waters.

L49 : you could mention here the study (46). Reference 46 was added in this sentence (Line 49).

L 58 : you could add here tropical latitudes (46). We have added tropical latitudes and the reference
(Line 58)

L84 and L198 : “excellent agreement” is a self-evaluation. Provide the stats of the comparison and let
the reader decide the quality of the agreement. The sentences on both lines were deleted (new Line 86
and Line 185)

Figure 2: it could be useful to add in this figure or as a separate supplemental figure the bathymetry
profile along these transects. Given these are alpine lakes, the bathymetry should be very steep, so
probably the littoral zone extremely restricted. This will not be case of other type of lakes.

The figure of the bathymetry profile along the transect (Fig. S4) was added to the SI as requested by the
reviewer and referenced in the caption of Fig 2.

L 128 : Fick’s law describes molecular diffusion and not air-water fluxes. K₆₀₀ does not have the
dimension of a diffusion coefficient (m²/s) but of a velocity (m/s)

Fick’s law describes diffusive fluxes, also across interfaces. At the air-water interface, it is Fick’s first
law, $F = k (C_{\text{sat}} - C_{\text{meas}}) / dz$ or $F = k dC/dz$. However, as dz cannot be resolved on the air-side or water-
side at the interface, k (in m²/s) is divided by dz (m) so $k_{600} = k/dz$ and Flux = $k_{600} (C_{\text{sat}} - C_{\text{meas}})$.

L 132: it’s usually called gas transfer velocity and not mass transfer velocity. This term has been called
in many ways, “gas transfer velocity” (Ulseth et al., 2019), “mass transfer coefficient” (de la Fuente et
al., 2015), “gas transfer coefficient” (MacIntyre, 2010). We choose to keep it as mass transfer
coefficient.

L 261: The eutrophic lakes BER and NOI also have much higher hypolimnetic CH₄ concentrations. So
the eutrophy promotes sediment CH₄. The reviewer is correct.

L 267: you could mention here the study (46). We have omitted this sentence.

L283: This was demonstrated before by (46) who reported a relation between OMP and primary
production. But in this case OMP was directly and accurately measured with stable isotope tracers
rather than computed indirectly by mass balance with a large uncertainty.

We agree with the reviewer about the relation between OMP and primary production, however in this
paper we propose that light regime also plays an important role on methane production.

Figure 6: Peeters and Hoffman (2021) recently showed that the estimates of Guenthel et al. (2019) and
Donis et al. (2017) were in fact wrong due to errors in terms of the mass balance. Are the values in this
graph the original (and erroneously over-estimated) values or the corrected values proposed by Peeters
and Hoffman (2021)?

These values we used are correct and as reported in Günthel et al. (2019). Please carefully read the
responses in full by Günthel et al. (2021) (also see our response on lines 406-415 on this document).
<https://www.nature.com/articles/s41467-021-21216-1>

The linear relation shown in Fig. S13 is not be statistically valid (specially given that the data have
been log transformed). With 8 data points a linear relation is valid at 0.05 level for r^2 higher than 0.5
which is not the case here ($r^2=0.37$). This figure is statistically meaningless and is not acceptable for a
journal such as Nature Comm. Figure should be removed, as well as the accompanying text.

We thank the reviewer for pointing this out. This was purely an exercise to test the validity of an OMP
mechanism offered by another study and does not change any of our results. After consideration, we
decide to keep the figure since this could be valuable information for future research. However we
remove where this was mentioned as a proposed upscaling approach (old discussion lines 388 and
Introduction line 90).

L 285: Statement “Methylphosphonate (MPn) biodegradation has been shown to be responsible for CH₄
production in oxic waters of the ocean (...)” is incorrect. What the Karl et al. (2008) showed is that if
you add MPn to oligotrophic seawater then CH₄ appears during the incubation. While this is an
interesting finding it’s hardly bullet-proof evidence of a process “responsible for CH₄ production in
oxic waters of the ocean”. The main objection being that MPn does not occur naturally, it’s an artificial
molecule for industrial applications.

Karl et al. (2008) amended seawater with MPn in their study and that MPn can be produced artificially;
however, there is evidence suggesting that it can be produced naturally in marine waters. Metcalf et al.
(2012) showed a marine archaeon (*Nitrosopumilus maritimus*) can synthesize MPn and that many other
marine organisms encode a pathway to also do so. They suggest that MPn is not readily measured in P-
limited marine environments because it is rapidly broken down. Repeta et al. (2016) found that marine

bacteria can produce methane using polysaccharide MPn esters in dissolved organic matter (DOM),
which is also a potential mechanism for oxic methane production in freshwaters with DOM. We have
thus reworded this opening sentence to better reflect the fact that this is a suggested mechanism to
explain the methane paradox (Lines 333-335).

L302: you could mention here the study (46). We already have three references here.

L304: light inhibition of MOX was directly quantified by (46) We have added ref 46.

The relation in Figure 6 are mainly driven by the data in the four alpine lakes reported here, the data
from the other lakes cluster close to the origin. So the relation mainly explains the data in 4 alpine
lakes. A relation that explains variations in 4 lakes hardly makes any progress “towards a global
scaling” as claimed in the title.

We think that this is an important step forward and is currently the best available approach, and now we
added five additional Canadian lakes to our upscaling approach from Thottathil et al., (2022) (new Fig
5b).

**Reviewer #2 (Remarks to the author):**

Ordóñez et al. present a very interesting dataset suggesting that oxic methane production occurs in the
studied lakes. The authors conclude this from the results of both a full-scale mass balance and a lateral
transport model to analyze sources and sinks of methane in the surface mixed layer of four pre-alpine
lakes in the Swiss Alps.

The study is well written, sections are generally clearly structured and the graphical presentation is
outstanding. Discussion and conclusions are refreshingly kept cautious but straight forward and focus
on the strengths and shortcomings of the used models and its parameters.

By this, I highly recommend this study for publication after revisions based on the comments, I listed
below.

**General comments:**

The study lacks any information about the calculation (or determination?) of methane oxidation. These
data are, however, mandatory for the calculation of PNet.

**Response:** We believe the reviewer missed an important aspect of our approach and that is that Pnet is
the net balance between OMP and methane oxidation. Thus, methane oxidation is not required to
calculate P_{net}. Basically, P_{net} is what we would need to close the mass balance in the SML ($P_{net} = F_a -$
$F_{sed} - F_z - R_{dis}$). Conceptually, P_{net} is the balance between OMP and methane oxidation ($P_{net} = OMP -$
MOx). Therefore, to calculate OMP it would be mandatory to calculate methane oxidation but not for
P_{net}. We have made this point clearer at the location where we initially addressed it in the manuscript
lines 73-82

Surface CH₄ concentrations were sampled along transects in each lake and by this, CH₄ concentration
was simulated using the lateral transport model. However, Figure 2 shows large deviations for near-
shore concentrations. These deviations should be mentioned clearly. Although PNet is calculated in the
center of the lake, the authors should state if & why the flux from the sediment is generally
underestimated by their full-lake model and how this affects the overall mass balance.

**Response:** We observe these anomalies in only four of twelve transects (BRE: June-18, NOI:Jul-19 and
Sept-18, and CHA: Jul-19). In all these cases, it was the measurement closest to shore that was
elevated. Near the shore in NOI and BRE, the growth of macroalgae can decrease the horizontal
dispersion coefficient and accumulate near-shore CH₄. This does not mean that the sediment flux was
under-estimated. For example, in BRE we measured one of the highest reported CH₄ sediment fluxes in
littoral sediment. In our calculation we do not include P_{net} spatial variability, and since it has been
observed that macroalgae can also produce CH₄ in oxic condition, P_{net} can be higher in the shore than in
the center of the lake. This phenomenon can result in an under-estimate in the modeled CH₄
concentrations near the shore. In the case of CHA, the littoral sediments are mostly rocks and the water
concentrations in the surface were low, we associate these differences to methodological issues or local
disturbance by people or cows (often standing in the water). This explanation was added to the text in
the lines 116-125

**Specific comments**

- • Line 111: Add SML in “along a transect from shore to shore”. Added in Line 113
- • Line 113: When was the difference in oversaturation significant between shore and center of the
lake? The significance was determined using ANOVA analysis (see caption Table 1).

- • Fig 2: add information about outliers in the caption. We added information in the main text. See
response in lines 612- 623 on this document about the outliers.
- • Line 128-148: Consider to move method specific information to material & methods section
This section was moved to the Methods section (Lines 506-510)
- • Line 163-165: Consider to move method specific information to material & methods section.
We keep the old lines 163-165 in the Results section because we think that gives information to
the reader about the terminology regarding the results. However we moved Line 165-167 to the
methods section (Lines 581-582).
- • Line 171-174: Consider to move method specific information to material & methods section.
Old lines 172-174 were deleted from the manuscript since it was already explained in the
Methods section.
- • Line 185-196: Consider to move method specific information to material & methods section.
This section was moved to the Methods section (Lines 433-454)
- • Line 239: Add information on trophic state. Since there is no relation between P_{net} contribution
(fractional CH_4 contribution to emissions of all sources) and trophic state (see discussion lines
253-263) we think this is not necessary to add this information in the result section. However
the trophic state information was added on the Figure 4.
- • Fig 5: Add “eutrophic” and “oligotrophic” similar to Fig 3. Trophic state information was added
to new Fig 4 (old Fig. 5).
- • Consider calculating R^2 for oligotrophic and eutrophic individually. Is there any significant
difference?
- We assumed that the reviewer was referring to the data in Fig 5 (new Fig 4.). But since there is
no relation between P_{net} contribution and trophic state we did not include this in the manuscript.
- • Line 309: Refer to Fig 6. We added the cross-reference to new Fig. 5B (new line 376)
- • Fig 7: Add “eutrophic” and “oligotrophic” similar to Fig 3.
We moved Fig 7 to the SI (new Fig. S12). Trophic state information was added to Fig. S12.
- • Line 475: add information on injected volume.

To measure CH₄ concentration in the water column we transferred the headspace into a 1 L gas
sampling bags (Supel Inert Multi-Layer Foil). Then the gas was directly measured to the Cavity
Ring-Down Spectrometer analyzer (Picarro G2201-i) until we observed a constant signal from
the instrument (about 100 mL of gas). Therefore we do not have any injected volume to report.
See Donis *et al.* (2016) for further details.

- • Line 476: correct to Picarro G2201-i. We had corrected to G2201-i (line 490)

**Reviewer # 3 (Remarks to the Author):**

**General comments**

The subject of the manuscript is relevant since oxic methane production is a little-known process, and
its investigation may help reduce uncertainties in large-scale methane flux budgets. The introduction is
well written and reflects the importance of the subject. The methods used are suitable, and authors have
clearly put in a lot of effort in producing a valuable and extensive dataset. Overall, this is a nice and
important study. However, I believe some major aspects of the paper need to be improved prior to
publication, mainly concerning the structure, interpretation, and focus of the discussion:

- • In terms of structure, the “Results” and “Methods” section are very confusing since a large
portion of the methodological information (including major equations) is found in the “Results”
section rather than the methods, making the manuscript hard to follow. Methodological and
results information are also found in the discussion section (ex. Line 321-334). Authors should
work on properly separating sections and avoid mixing the different types of information.
**Response:** Following the suggestions of reviewer #2 and #3 we decided to move the Lines 128-
148, 165-167, 185-196 to the Methods section. However, some methodological part were kept
in the Results and Discussion section to help the reader to follow the main ideas of each
paragraph (Lines 272-288).
- • The discussion section is largely focused on justifying the validity of the Pnet estimates as and
contrasting it to Peeters et al. study. While justifying methods and backing up observed results
is important, it should not be the main focus of the discussion. Despite the original paradigm
and debate associated with oxic methane production, many recent studies have shown, using
different approaches, the existence and importance of this process, and it is now an established

methane source. Authors could instead expand the discussion on potential OMP drivers, which
is rather limited in the manuscript. For instance, further explore results, analyses, and
hypotheses on seasonal and inter-lake patterns of OMP rate and contribution to flux.

**Response:** We thank the reviewer for his comments. We now have expanded the discussion
about the role of LC as a main driver of P_{net} in stratified lakes (Lines 354-365 in the manuscript,
and described more in detail in lines 366-401 on this document). We have also largely removed
mention of the Peeters et al. paper in the discussion. About the differences between lakes, we
stated these could be related to the different trophic states that in our case are represented by
surface CH_4 concentrations, Chla and light conditions (Lines 366-379). Despite these
differences, we also observed that the seasonality can be explained by LC (new Fig 5a).
However, we think that it is not possible to deeply discuss the seasonal trends given the limited
dataset on the temporal scale (only three points).

• In the discussion, authors mention that uncertainty on Pnet estimates are largely related to
uncertainty in K_{600} . This is in itself a novel and important result for this study and future
assessments of OMP. However, results of the uncertainty calculation (sensitivity analysis) are
not clearly shown. It would be useful to mention in the results / figures / discussion what is the
actual change in OMP caused by a certain error on k_{600} . This is especially important given that
both measured and modeled k_{600} values have associated error. Indeed, measured k_{600} is
restricted to a 5-min time scale, not necessarily representative of diel patterns. This limitation of
measured k_{600} should also be discussed in the choice of method and interpretation of the date.
In terms of OMP uncertainty it would also be interesting to present values on the sensitivity of
OMP estimates to factors other than k_{600} as well for comparison. This would yield novel and
constructive information in terms of advancing OMP estimation in a larger context.

Thank you for this comment. We agree that this is an important conclusion for the paper. For the
uncertainty calculations, these are stated in lines 193-218. We discuss the limitations of K_{600} in
lines 278-296. We also refer you to our response to R1 on the use of floating chambers (lines
296-315 in this manuscript).

• The manuscript contains 7 figures and 3 tables in the main text, with an additional 14 figures
and 5 tables in the supplementary information. This number of displays is unreasonable, and
reflects a general lack of structure in the study, failing to summarize and focus on important

information and target specific research questions (especially in the discussion). Authors should
drastically reduce the number of displays based on the main messages (conclusions) they want
to convey.

We have made the main text as concise and focused as possible. We currently have 5 figures and
3 tables. We believe that the figures in the supplemental provide a lot of background data and
information for researchers who wish to have access to these data or reproduce our results. We
defer to the editor's input as to whether the amount of figures in the supplemental is excessive.

**Specific comments**

- • Isotopic signature data are presented in the results but never mentioned in the discussion. Thus,
they should be either removed from the manuscript or interpreted in the discussion if they
provide valuable insights.
- • Line 12: more widely accepted than what? Be specific. We reformulated the sentence to (Line
12): “Oxic methane production (OMP) in freshwater is widely accepted, ...”
- • Line 20: at this point it is not „newly recognized“ given historic marine research. Revise. We
reformulated the sentence to (Line 20): “OMP is a direct CH₄ source ..”
- • Line 22: This statement is alarmist and must be changed, A positive feedback loop assumes that
oxic lake CH₄ emissions will deliver enough CH₄ due to eutrophication/climate change (above
and beyond baseline oxic fluxes) that it significantly drives the atmospheric CH₄ concentration
up. The scale of the flux we are talking about here is nowhere near that large, and the
mechanistic evidence for the statement is not available. Maybe refocus conclusion sentence to
emphasize the importance of the dataset and modelling technique and how it advances
understanding of the lake CH₄ cycle. We agree with the reviewer that it is premature to consider
a potential positive loop between climate change and OMP, however, we respectfully disagree
with the reviewer about the small contribution of lakes on the global lake budget. It has been
shown that methane contributes about 50% of the natural CH₄ emissions (Rosentreter et al.,
2021). Despite that OMP contribution varies between lakes and time of the year, OMP is an
important source of methane contribution up to 80% (Donis et al., 2017, Günthel et al., 2019).

- Therefore, line 22 was changed to: “... specially the potential links between climate change,
phytoplankton production, OMP and CH₄ emissions to the atmosphere” (new lines 22-23)
- • Line 33: since onset of industrial era? We added the word “onset” (Line 33)
 - • Line 43: it is quickly. We reformulated to: “can be quickly emitted” (Line 43)
 - • Line 58: altitude We have changed to altitude (Line 58)
 - • Line 71: “with identical climate” change “with” to “facing”. We reformulated to: “under
identical climate (Line 72)
 - • In the description of the model (ex. Line 72 - 74), the formulation sound like OMP is the only
methane source considered in the model. Please clarify that other sources are accounted and
OMP is not the only process “that adds CH₄” (line 73). Conceptually, P_{net} is the balance
between OMP and MOx in the SML. But to calculate P_{net} we used all the sources and sinks in
the SML (see lines 73-82, Eqns. 1 and 2). We reformulate the sentence: “The net CH₄
production rate (P_{net} , Fig. 1) is defined as the balance between OMP (that adds CH₄) and CH₄
oxidation (MOx, that removes CH₄) from the surface mixed layer (SML)^{3,37,73}. P_{net} in the SML
was estimated using two independent mass balance approaches: a 0-D full-scale mass balance
following Donis et al.¹⁷ and a 1-D lateral transport model adapted from to Peeters et al.⁵⁶ where
...” (lines 73-78)
 - • Line 85: add the actual numbers for the contribution of OMP. We reformulated to: “OMP
contributes between 30 to 90 %.” (line 87)
 - • Line 113: Don’t agree with this artificial breakdown without some proof up front that each lake
falls in the trophic status categories they are assigned to. This applies here and throughout. The
trophic state index (TSI) based on total phosphorus (TP), Secchi depth (SD) and chlorophyll-a
was calculated based on Carlson Trophic State Index (Carlson et al. 1977) to classify the trophic
state of each lake based the June 2018 data. This table was added to the SI (Table S2). This
table was referenced in Line 104.

Lake	TSI(TP)	TSI(SD)	TSI(Chla)	Avg. TSI	Classification
Bretaye	55	41	65	54 ± 12	Eutrophic
Noir	36	45	65	49 ± 15	Eutro/Mesotrophic
Chavonnes	34	38	58	44 ± 13	Mesotrophic
Lioson	26	28	59	38 ± 18	Oligotrophic

- • Line 121: color in this graph and others is not distinct enough in greyscale. The color scales of
the figures were selected for color blind people. We would prefer to leave them as is.
- • Line 131-132: add reference for the exponent. Reference to McGinnis et al 2015 and Vachon et
al 2010 were added (Line 510)
- • Line 148: Hypereutrophic? How is this possible? Was changed to Eutrophic. See response for
trophic state classification on lines 759-764 on this document.
- • Line 152: less enriched, not “lighter” – use proper terminology. Line 152 was reformulated as:
“Littoral sediment was around 20% isotopically less enriched than the surface water of NOI and
BRE ...” (line 154)
- • Line 200-203: the values in the two sentences do not seem to match. These are the average P_{net}
rate of the Eutrophic lakes (NOI and BRE) and the oligotrophic lakes CHA and LIO was about
890 and 245 $\mu\text{molm}^{-3}\text{d}^{-1}$ respectively (about 7 times lower). However, we reformulate the
sentence to make it clear that it is the average of NOI and BRE against CHA and LIO together.
Line 189-190: “On average, P_{net} rates in eutrophic lakes (BRE and NOI) were about seven
779 times higher than the oligotrophic lakes (CHA and LIO).”
- • Line 230: But I count 6 points below the 1:1 line in fig. 4c. Is this statement correct? Yes, it is
correct. We calculated the mean normalized bias (MNB) for the data on Fig S9c (old Fig 4c),
and it shows that in this case the model underestimates the observations (MNB=-0.36, Table 2.).
- • Line 245: Is the comparison in fig 4b showing a circular relationship? Is the chamber-based flux
contained in both x and y axis metrics? Confirm this is not a problem with a quick sensitivity
analysis?
- In Fig S9b (old Fig 4b) the y-axis is the measured surface concentration and the x-axis is the
simulated concentration with the lateral transport model using the results from the full-scale
mass balance approach. We do not think this is a circular relation because the measured
concentration is an independent variable, and also because the full-scale mass balance does not
use the measured surface concentration to estimate P_{net} .
- • Line 251-253: is there a link between P_{net} and water level? If we split the analysis between
shallow (BRE and NOI) and deep lakes (CHA and LIO) we observed that P_{net} is higher in
shallower lakes. However, this relation would not explain why P_{net} changed in time (eg in BRE

- and NOI) without any change in water level. Also if we analyze CHA and LIO we observe the
opposite relationship, i.e., higher P_{net} in the deeper lake. Therefore, we do not think that water
level can play a role on P_{net} rates.
- • Line 256: this is a weird sentence formulation. Within or across sites? Changed to “ P_{net} rates
were temporally variable in each lake and also varied between study sites”. (line 242)
 - • Line 257 to 265: Since P_{net} and MOx not directly confirmed here, please be more cautious in
these statements as the mass-balance essentially remains unvalidated. We changed this section
and replaced OMP with P_{net} , since that is what we resolved.
 - • Line 267: Disagree, it is not hotly debated. One group disagrees, but the vast majority of papers
show some kind of support and converge to an agreement that OMP cycling is relevant to
surface water processes. Old line 267 (new 253) was changed to “The two dominant sources of
CH_4 to the surface waters are P_{net} ^{17,23,74} and CH_4 flux from littoral sediment^{18,52}.”
 - • Line 274 to 276: Seems like an obvious statement that could be replaced with a more
informative conclusion. We have reformulated to: “our results suggest that there is no
relationship between the contribution of the two dominant CH_4 sources (P_{net} and F_s) and trophic
state, even though each of these sources are higher in more productive systems” (lines 261-263)
 - • Line 286 to 288: Wouldn't P limitation depend more on N to P ratios, not only [P]? We agree
with the reviewer that P limitation depends more on the N:P ratios rather than P concentrations.
However, P_{net} does not correlate with N:P ratios ($R^2=0.04$, figure not shown). While OMP rates
have been related with P and N limited environments, we only include the analysis between P_{net}
versus P and N.
 - • Line 294: Also this relationship is shown in reference 6. Could cite as further support.
Reference 6 was added. (line 343)
 - • Line 299-301: I don't quite understand this approach, please clarify. Please read our next
response
 - • Line 305-307: I don't understand the choice of these parameters. Why include both Chla and
light parameters if light only influences P_{net} indirectly through Chla? Also, this statistical
analysis is not described in the methods or results, it is a multiple regression? What is the
equation? Is it presented in a figure?

We include both Chla and light parameters because we observed for each lake a linear
correlation between P_{net} and LC (Fig 5a). Because this relationship changes depending on the
trophic state, we have included Chla, CH_4 and SD as trophic parameters to explain the
differences between lakes. With our approach, we hypothesized that Chla will produce CH_4
under oxic conditions only when they are exposed to high intensity light. For further details,
please read our response on lines 366-401 on this manuscript. About the equation please read of
following response.

• Line 309-312: I disagree with the authors that this particular result provides “a step towards
estimating P_{net} ” as they provide no numerical equation, and the link between OMP and Chla is
already known.

We respectfully disagree with the reviewer because the equation was provided inside Fig 6 (now
Fig 5b) where X correspond to $Chla \times LC \times SD$. While the reviewer is correct when he states
that the relation between OMP and Chla was already known, there is no approach to upscale this
relation to natural systems. Moreover, we proposed in our upscaling approach that light
conditions are also an important driver to estimate P_{net} in different systems.

• Line 333 to 339: Rethink this interpretation? Day-time chambers are selecting for the most
turbulent time of day, so it makes sense that other models will have a lower (possibly more
representative) k_{600} ?

During the day turbulence in surface waters is mainly driven by wind but can be suppressed by
stratification due to warming. However during the night, surface convection increases by
cooling the surface water. These lakes are surrounded by mountains that protect them from high
wind exposure and, on the other hand, they experience high surface cooling during the night,
increasing turbulence. Although research has to be conducted to compare day and nighttime
flux chambers measurements in high altitude lakes, we think that day-time chambers were not
necessarily conducted during the most turbulent time of day.

• Line 350: Does this take into consideration the fact that day-time measurements don't account
for night-time physical conditions that are often quite different?

We thank the reviewer for this comment. As explained in the previous response, the conditions
during day and night are different, but not necessarily mean that one is more turbulent than the

other in high altitude lakes. That is why we recommend using measured values rather than k_{600}
parameterizations.

- • Line 381: “need it” should be “needed”?

We have corrected the wording to “necessary” (Line 388)

- • Line 421: the term mass balance is confusing here because it refers to both the full-scale mass
balance and the lateral transport model if I understand correctly?

Both approaches are based on the mass balance principle but they have different assumptions.
The full-scale mass balance (0-D) assumes that the SML is a well-mixed reactor, while the
lateral transport model (1-D) assumes that the boundary conditions and the CH₄ concentration
change in space. Therefore, P_{net} is calculated based on two mass balance approaches that share
the same principle but different assumptions. Since the changes suggested moving the
methodological parts written in the results section the mass balance description in the method
section was rewritten (Lines 430-469)

- • Line 438: “horizontal” should be “horizontal plane”? Yes, we had changed to horizontal plane
(Line 456)

- • Line 478: specify what is the headspace/water ratio used. We have added the “headspace/water
ration (500 mL air/ 500 mL water)” (Line 493)

- • Table S2: showing means and SD or medians may be more informative than just the range here.
The only range shown in Table S2 (new Table S6) is in the Depth column. As described in the
Method section, nutrients were obtained from an integrated sample technique. The Depth
column shows the range of the integrated sample, and the values in the nutrients are the average
of triplicates. This description was added to the caption of Table S6 “The Depth column
describe the range of depths from/to the samples were integrated (Method)”

- • Table S2: It is weird that the dates are not shown in order (Sep. before July). The order was
changed in Table S6 (old Table S2) to July before September

**References**

Cole, Jonathan J, Bade, Darren L., Bastviken, David, Pace, Michael L., Van de Bogert, Matthew,
(2010), Multiple approaches to estimating air-water gas exchange in small lakes, *Limnol. Oceanogr.*
*Methods*, 8, doi:10.4319/lom.2010.8.285.

Cory J. D. Matthews, Vincent L. St.Louis, and Raymond H. Comparison of Three Techniques Used To
Measure Diffusive Gas Exchange from Sheltered Aquatic Surfaces. *Hesslein Environmental Science &*
*Technology* 2003, 37 (4), 772-780 DOI: 10.1021/es0205838

DelSontro, T., Boutet, L., St-pierre, A., Prairie, Y.T.. Methane ebullition and diffusion from northern
ponds and lakes regulated by the interaction between temperature and system productivity. *Limnology*
*and Oceanography* 2016, 61, S62–S77. <https://doi.org/10.1002/lno.10335>

Ernst, Leonard, Benedikt Steinfeld, Uladzimir Barayeu, Thomas Klintzsch, Markus Kurth, Dirk
Grimm, Tobias P. Dick, Johannes G. Rebelein, Ilka B. Bischofs, and Frank Keppler. “Methane
Formation Driven by Reactive Oxygen Species across All Living Organisms.” *Nature*, March 9, 2022,
1–6. <https://doi.org/10.1038/s41586-022-04511-9>.

Lorke, Andreas, Pascal Bodmer, C Noss, Zeyad Alshboul, M Koschorreck, David Bastviken, Sabine
Flury, D. F. Mcginnis, Andreas Maeck, and Katrin Premke. “Technical Note: Drifting versus Anchored
Flux Chambers for Measuring Greenhouse Gas Emissions from Running Waters.” *Biogeosciences* 12
(2015): 7013–24. <https://doi.org/10.5194/bg-12-7013-2015>.

Perolo, Pascal, Bieito Fernández Castro, Nicolas Escoffier, Thibault Lambert, Damien Bouffard, and
Marie-Elodie Perga. “Accounting for Surface Waves Improves Gas Flux Estimation at High Wind
Speed in a Large Lake.” *Earth System Dynamics* 12, no. 4 (November 16, 2021): 1169–89.
<https://doi.org/10.5194/esd-12-1169-2021>.

Schubert, Carsten J., Torsten Diem, and Werner Eugster. “Methane Emissions from a Small Wind
Shielded Lake Determined by Eddy Covariance, Flux Chambers, Anchored Funnels, and Boundary
Model Calculations: A Comparison.” *Environmental Science & Technology* 46, no. 8 (April 17, 2012):
4515–22. <https://doi.org/10.1021/es203465x>.

Reviewer #2 (Remarks to the Author):

All points raised have been addressed and all significant changes have been made in the new version of the manuscript. I now recommend this paper to accept as is.

Reviewer #4 (Remarks to the Author):

Please see the attached report.

Reviewer #4 Attachment on the following page.

Comments on manuscript entitled “*Towards a global scaling of oxidic methane production: evaluation of the methane paradox in four Swiss pre-alpine lakes*” by Ordóñez et al.

The authors have put a good effort in answering the comments from the reviewers increasing the quality of the manuscript. The addressed most of the comments in a satisfactory manner. However, I still think some improvement should be made to make the manuscript publication ready:

- While the authors spend a large part of the discussion on the comparison between measured and modeled K_{600} , they do not actually get into the impact of K_{600} on P_{net} estimates. The question that should be addressed here is : if K_{600} is over or under estimated by for instance 50%, by how much would P_{net} be over/under estimated. I think this information is important first because it would make the P_{net} uncertainty much more transparent to the reader, and second because it would guide future research on the actual (quantified) importance K_{600} on P_{net} estimates. In this regard I reiterate my suggestion of doing sensitivity analyses to numerically show the effect of K_{600} on P_{net} .
- Related to the previous comment, the authors have not discussed the uncertainty related to using a K_{600} measured within a very short time range (a few minutes) as representative of a daily value. While I agree with the authors that a K_{600} derived from chamber measurements is the best approach, I also think the uncertainties related to this approach should be discussed and shown. This could complement well the sensitivity analysis suggested in the previous comment.
- While the relationship in Fig. 5 is interesting, I think talking about a global relationship is a bit of an overstatement since the number of lakes is still very limited and nowhere near an appropriate spatial distribution required for a global trend. I don't mind the authors talking about the study being a step forward an eventual global-scale understanding of OMP, however, I don't think the work 'global' belongs in the title.
- While the authors have reduced the number of figures and tables in the manuscript, I think they could easily reduce the figures in the supplementary information too by combining them which would be more efficient and more pleasant to read. While the authors say that the decision on the number of figures ultimately goes to the editor, I think they should aim at making their paper as easy to read and understand as possible, which is not the case with these many figures.

**Reviewer 4**

The authors have put a good effort in answering the comments from the reviewers increasing the
quality of the manuscript. The addressed most of the comments in a satisfactory manner. However, I
still think some improvement should be made to make the manuscript publication ready:

- • While the authors spend a large part of the discussion on the comparison between measured and
modeled K600, they do not actually get into the impact of K600 on Pnet estimates. The question
that should be addressed here is: if K600 is over or under estimated by for instance 50%, by
how much would Pnet be over/under estimated. I think this information is important first
because it would make the Pnet uncertainty much more transparent to the reader, and second
because it would guide future research on the actual (quantified) importance K600 on Pnet
estimates. In this regard I reiterate my suggestion of doing sensitivity analyses to numerically
show the effect of K600 on Pnet.

We have included a new sensitivity analysis where we estimated the error caused in the Pnet
calculation based on an over- or underestimation of k_{CH_4} caused by using k_{600} literature
parameterizations. The percentage of the Pnet error (P_{net}^{err}) depends on the ratio between the measured
flux to the atmosphere and the Pnet rates estimated from measured values (see Methods lines: 484-
489). Excluding the cases when Pnet was negligible in the sampled lakes, this ratio was between 0.5
and 3 with an average of 1.3, meaning that for a given percentage error of k_{CH_4} the Pnet error will be on
average 30% higher than the error of k_{CH_4} (see figure below). These results highlights the need to
accurately estimate k_{600} to conduct CH_4 mass balances in the surface layer of lakes. This analysis is
now included in the discussion section (Lines: 272-279)

Figure S17. Percentage of error on P_{net} calculations ($P_{\text{net}}^{\text{err}}$) caused by over- or underestimation of k_{CH_4}
 (k_{err}) due to the use of k_{600} parameterizations. Estimating the ratio between $F_a A_a$ and $P_{\text{net}} V_{\text{SML}}$ from
 measured values (lines), it is possible to estimate the P_{net} error given the percentage error on k_{CH_4}
 estimations.

- • Related to the previous comment, the authors have not discussed the uncertainty related to using
 a K_{600} measured within a very short time range (a few minutes) as representative of a daily
 value. While I agree with the authors that a K_{600} derived from chamber measurements is the
 best approach, I also think the uncertainties related to this approach should be discussed and
 shown. This could complement well the sensitivity analysis suggested in the previous comment.

We agree with the reviewer that short time measurement of surface fluxes might be not representative
 of a daily average value. In our analysis, we used about 10 chamber measurements along a transect and
 five measurements at the center of the lake to calculate the average flux values during the day (Lines
 116-118). We have mentioned this in the discussion (Lines: 250-252).

- • While the relationship in Fig. 5 is interesting, I think talking about a global relationship is a bit
 of an overstatement since the number of lakes is still very limited and nowhere near an
 appropriate spatial distribution required for a global trend. I don't mind the authors talking

about the study being a step forward an eventual global-scale understanding of OMP, however, I
don't think the work 'global' belongs in the title.

We agree with the reviewer and have changed the title to "Evaluation of the methane paradox in four
Swiss pre-alpine lakes across a trophic gradient: towards an up-scaling approach" We made this change
to highlight that we are proposing a new approach to estimate OMP based on physicochemical
parameters (Secchi depth, Chla concentrations and surface mixed layer depth) that could potentially be
used a first order up-scaling method.

• While the authors have reduced the number of figures and tables in the manuscript, I think they
could easily reduce the figures in the supplementary information too by combining them which
would be more efficient and more pleasant to read. While the authors say that the decision on
the number of figures ultimately goes to the editor, I think they should aim at making their
paper as easy to read and understand as possible, which is not the case with these many figures.

We thank the reviewer for this comment. We, of course, want the paper to be easily digestible by the
reader; therefore, we have combined former Fig S13 and Fig S14 into one figure (now Fig S13).
Unfortunately, we could not find any other figures to cut without losing important information or any
other figures to combine without making highly complex figures/panels that would be difficult to
follow. Ultimately, we had to add one more figure for the k600 analysis (as requested by the reviewer),
and thus the number of figures remains the same. We believe that the figures in the supplemental
provide necessary background data and information for researchers who wish to have access to these
data or reproduce our results. We defer to the editor's decision as to whether the number of figures in
the supplemental is excessive.